# Binary dopant segregation enables hematite-based heterostructures for highly efficient solar $H_2O_2$ synthesis

Zhujun Zhang[1], Takashi Tsuchimochi[2,3], Toshiaki Ina[4], Yoshitaka Kumabe[1], Shunsuke Muto [5], Koji Ohara [4], Hiroki Yamada[4], Seiichiro L. Ten-no[2,6] & Takashi Tachikawa [1,7 ✉]

Dopant segregation, frequently observed in ionic oxides, is useful for engineering materials and devices. However, due to the poor driving force for ion migration and/or the presence of substantial grain boundaries, dopants are mostly confined within a nanoscale region. Herein, we demonstrate that core–shell heterostructures are formed by oriented self-segregation using one-step thermal annealing of metal-doped hematite mesocrystals at relatively low temperatures in air. The sintering of highly ordered interfaces between the nanocrystal subunits inside the mesocrystal eliminates grain boundaries, leaving numerous oxygen vacancies in the bulk. This results in the efficient segregation of dopants (~90%) on the external surface, which forms their oxide overlayers. The optimized photoanode based on hematite mesocrystals with oxide overlayers containing Sn and Ti dopants realises high activity (~0.8 μmol min$^{-1}$ cm$^{-2}$) and selectivity (~90%) for photoelectrochemical $H_2O_2$ production, which provides a wide range of application for the proposed concept.

[1] Molecular Photoscience Research Center, Kobe University, 1-1 Rokkodai-Cho, Nada-Ku, Kobe 657-8501, Japan. [2] Graduate School of System Informatics, Kobe University, 1-1 Rokkodai-Cho, Nada-Ku, Kobe 657-8501, Japan. [3] PRESTO, Japan Science and Technology Agency (JST), 4-1-8 Honcho Kawaguchi, Saitama 332-0012, Japan. [4] Japan Synchrotron Radiation Research Institute, 1-1-1 Kouto, Sayo-Cho, Sayo-Gun, Hyogo 679-5198, Japan. [5] Electron Nanoscopy Section, Advanced Measurement Technology Center, Institute of Materials and Systems for Sustainability, Nagoya University, Furo-Cho, Chikusa-Ku, Nagoya 464-8603, Japan. [6] Graduate School of Science, Technology, and Innovation, Kobe University, 1-1 Rokkodai-Cho, Nada-Ku, Kobe 657-8501, Japan. [7] Department of Chemistry, Graduate School of Science, Kobe University, 1-1 Rokkodai-Cho, Nada-Ku, Kobe 657-8501, Japan. ✉email: tachikawa@port.kobe-u.ac.jp

Ionic oxide heterostructures have attracted significant attention in diverse fields ranging from catalysis to (magneto) optoelectronics owing to their tunable optical, electrical, and magnetic properties by precisely controlling the concentration and location of elements[1–3]. These heterostructures are mostly fabricated by vacuum technologies, such as atomic layer deposition[4] and chemical vapor deposition[5]. They rely on high-precision equipment and specialized precursor reagents, which limit their large-scale application for production in industries.

Dopant segregation is another approach. It usually occurs in ionic solids containing aliovalent dopant ions and is driven by elastic and/or electrostatic interactions (Fig. 1a)[6–8]. However, in many cases, only small amounts of dopants can reach the external surface of polycrystalline or nanocrystalline materials under elevated temperatures (i.e., 1300 °C for Sn-doped hematite[9]) owing to the limited ion migration that results from poor driving forces or grain boundaries (GBs) (Fig. 1b)[10]. Extrinsic or intrinsic defects (e.g., vacancies and interstitial atoms) in the crystals yield space charge regions that modify a local electrostatic potential,

but often lead to inhomogeneous properties and inevitably degrade their performance[11]. Thus, it is challenging to build the heterostructures by dopant segregation. External segregation may be promoted by removing the GBs from doped materials and adding excess space charges; however, these actions are incompatible. The concept of mesocrystal (MC)[12,13], which is an ordered assembly of nanocrystals via oriented attachment, provides a solution to this problem. We recently discovered that thermal treatment at relatively lower temperatures (e.g., 700 °C for hematite ($\alpha$-Fe$_2$O$_3$) MCs) induces the sintering of an interface (i.e., GB elimination) and creates numerous interfacial oxygen vacancies ($V_O$)[14,15], which facilitate the oriented migration of dopant ions as well as photogenerated charges (Fig. 1b).

Charge transfer efficiency and catalytic activity of semiconducting materials are highly influenced by their bulk electronic and surface structures. For example, hematite MCs with thin rutile TiO$_2$ overlayers exhibit excellent performance in photoelectrochemical (PEC) water oxidation to obtain O$_2$ owing to the suppressed surface recombination of photogenerated electrons and holes[14,15]. H$_2$O$_2$,

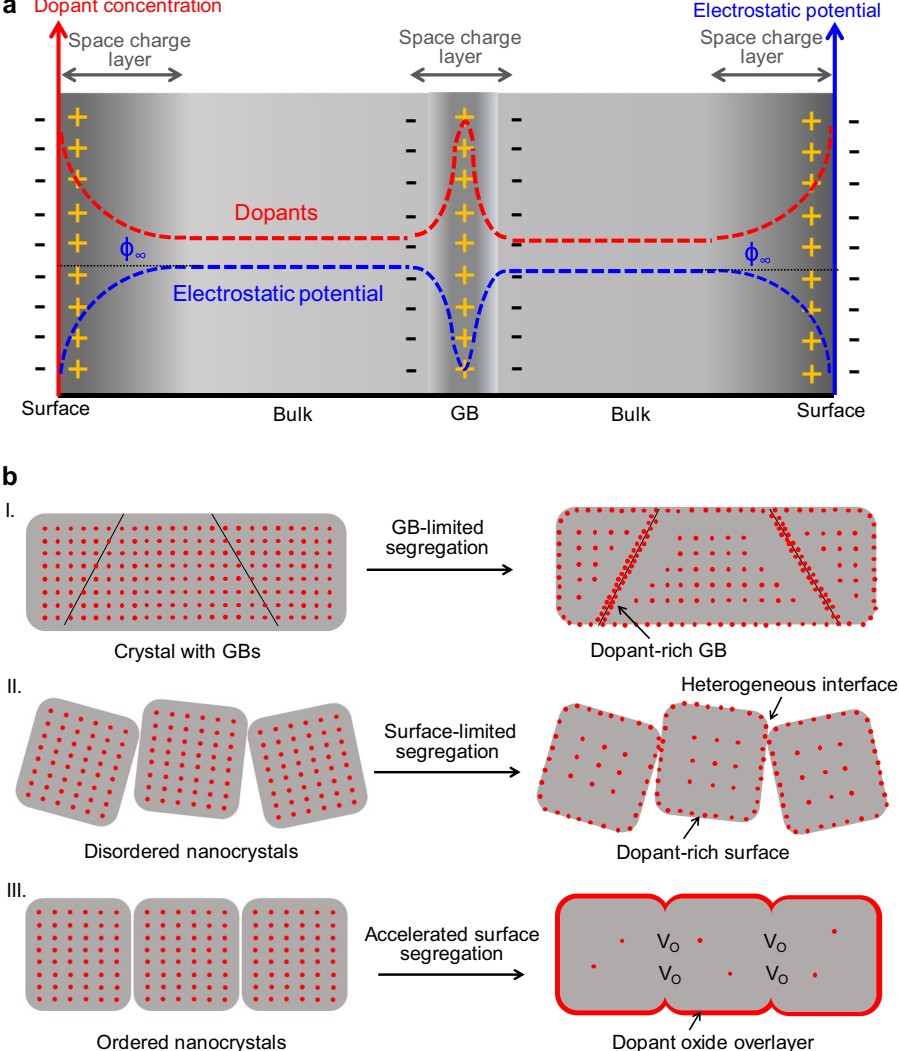

**Fig. 1 Space charge-induced dopant segregation. a** Distribution of dopants and electrostatic potential (donor-doping case) in ionic oxides based on the space charge theory (Supplementary Fig. 1 for acceptor-doping case). Strong segregation of dopants in the space charge layers is induced to compensate the excess charges on the crystal surfaces and bulk GBs (Supplementary Fig. 2). See Supplementary Note 1 for more details on the space charge theory. **b** Schematic illustration of the dopant segregation in different types of ionic oxide crystals: (I) Crystal with numerous GBs: dopants tend to segregate at the surface and GBs. (II) Disordered nanocrystals: a small number of dopants tend to segregate on the surface due to the lack of driving force even at high temperatures. (III) Ordered nanocrystals with highly aligned interfaces: a large number of dopants segregate on the outer surface because of interface sintering (GB elimination), creating numerous interfacial V$_O$ and narrowing the space charge layer to drive the charge migration.

which is another product of the water oxidation, exceeds $O_2$ owing to its usefulness as a green oxidant for industrial chemistry and environmental purification, as well as a clean energy source for fuel cells[16,17]. The PEC $H_2O_2$ production has been mostly realized by a two-electron pathway from water oxidation using $BiVO_4$-based photoanodes[18-21]; however, these photoanodes are still unstable for practical use due to the dissolution of $V^{5+}$ that arises from anodic photocorrosion[22].

Herein, we present MC-based dopant segregation to easily and effectively modify the surface of hematite MCs for highly efficient and selective solar-driven $H_2O_2$ production. Hematite (α-$Fe_2O_3$) is naturally abundant with good stability and suitable bandgap (~2.1 eV) for sunlight absorption and has been extensively studied for solar-driven water oxidation to obtain $O_2$[23,24]. To the best of our knowledge, there are no reports on hematite-based photoanodes for water oxidation to obtain $H_2O_2$, probably due to the unfavorable surface properties required for $H_2O_2$ generation[25-27]. By controlling the type and concentration of the dopants, as well as the annealing conditions, the hierarchical structures and catalytic activities of hematite MCs can be rationally optimized.

## Results

**Structures of hematite-based MCs.** The as-synthesized Sn, Ti-codoped $Fe_2O_3$ (SnTi–$Fe_2O_3$) MCs (Supplementary Fig. 3), which cause self-segregation during heating, exhibit uniform cuboid-like morphology (length = ~150 nm, width = ~110 nm, and height = ~90 nm) assembled with closely stacked nanocrystal subunits (~20 nm), as observed from the field-emission scanning electron microscopy (FE-SEM) (Supplementary Fig. 4) and transmission electron microscopy (TEM) images (Fig. 2a, Supplementary Fig. 5). A coherent crystal lattice of hematite (104) is seen with adjacent nanocrystals and intimate interfaces (Fig. 2b and Supplementary Fig. 5c), indicating that the as-synthesized particle is composed of crystallographically aligned nanocrystals. The specific structure of the MC is further featured by single crystal-like diffraction spots obtained from the adjacent nanocrystals (Supplementary Fig. 5d)[28,29]. After annealing it at 700 °C in air for 20 min, GBs in the bulk region almost disappeared, while disordered overlayers with a thickness of 1–7 nm were formed on the outer surface and inner mesopores were created inside the crystal (Fig. 2c, d). The interfacial sintering also creates numerous interfacial $V_O$, which appear as deficient regions in the crystal (white dotted line in Fig. 2c), resulting in an exceedingly high carrier density ($10^{20}$–$10^{21}$ cm$^{-3}$) and ultrathin space charge layers (Supplementary Note 2 and Supplementary Table 1)[14,15]. The energy-dispersive X-ray (EDX) maps (Fig. 2d) and corresponding EDX spectra (Supplementary Fig. 6) suggest that the concentrations of Ti and Sn are higher at the surface region than in the bulk. We obtained hematite MCs with dopant oxide overlayers ($SnO_2$ or $TiO_2$) by the same procedure (Supplementary Fig. 7), confirming the viability and validity of our approach.

We then employed high-angle annular dark-field scanning TEM (HAADF-STEM) combined with electron energy loss spectroscopy (EELS) to derive information about the local compositions, structures, and chemical states of the annealed SnTi–$Fe_2O_3$ MCs. Ti and Sn species are present with the same spatial distribution at the outer surface and near the edge of pores (as specified by the white dotted circles) (Fig. 2e), implying a single phase of the Sn–Ti complex. This speculation was partly supported by the fact that aligned high-loss EEL spectra obtained from the surface region show much higher peak intensities of Ti-$L_{2,3}$ and Sn-$M_{4,5}$ signals (Supplementary Fig. 8). The thermal accelerated dopant segregation at the crystal surface was further suggested by the significantly increased concentrations of the

dopants at the surface region based on the X-ray photoelectron spectroscopy (XPS) depth profiles (Fig. 2f). In addition, the XPS analysis revealed that the valance state of Sn ions for the annealed SnTi–$Fe_2O_3$ is lower than that ($Sn^{4+}$) of the annealed Sn–$Fe_2O_3$ (Fig. 2g). This was further confirmed by the Sn K-edge X-ray absorption near edge structure (XANES) spectrum of the annealed SnTi–$Fe_2O_3$ measured in a conversion electron yield (CEY) mode, an effective method to analyze the surface compositions of materials, which indicates a clear shift to lower energy compared with Sn–$Fe_2O_3$ and $SnO_2$ (Fig. 2i). The corresponding Sn K-edge extended X-ray absorption fine structure FT spectrum of SnTi–$Fe_2O_3$ exhibits the main first shell of Sn–O at 1.38 Å and second Sn–Ti shell at 2.76 Å[30,31], in addition to the weak coordination of Sn–Sn as those of $SnO_2$ and Sn–$Fe_2O_3$ at ~3.28 Å. The above results indicate the formation of SnTiO$_x$ hetero-overlayer with the possibility of a small amount of $SnO_2$ at the outer surface, as suggested by the Sn 3$d$ XPS depth profile analysis (Supplementary Fig. 8d).

**Heterostructures formed by dopant segregation.** All doped hematite MCs are composed of a hematite phase, as deduced from the powder X-ray diffraction (XRD) patterns (Supplementary Fig. 9). However, owing to the lattice expansion via the replacement of smaller $Fe^{3+}$ ions (0.55 Å) with larger $Sn^{2+}$ (0.999 Å) and/or $Ti^{4+}$ (0.605 Å) ions[32,33], the peak positions of the hematite diffraction lines shifted towards smaller angles (solid lines in Fig. 3a). The corresponding lattice $d_{(104)}$ space values increased from 2.709 to 2.730 Å with an increase in total dopant concentration from 0 to 30 mol% (Supplementary Fig. 10), confirming the uniform incorporation of Sn and/or Ti ions in the hematite lattice owing to the replacement with $Fe^{3+}$ ions[34]. After thermal treatment, the diffraction peak positions (dashed lines in Fig. 3a) and $d$ values of the doped samples exhibited the same level of undoped $Fe_2O_3$ (Fig. 3b), suggesting that most of the doped $Sn^{2+}$ and/or $Ti^{4+}$ ions (~90%) were diffused out from the hematite lattice. We observed a broad diffraction peak located at 27.0° (and 26.5°) for the annealed Ti–$Fe_2O_3$ (and Sn–$Fe_2O_3$) sample (Fig. 3c), supporting the formation of the (110) phase of the $TiO_2$ (and $SnO_2$) overlayer. For the annealed SnTi–$Fe_2O_3$ sample, no clear diffraction peak was detected, probably due to the disordered structure of the overlayer, as indicated by the HRTEM image (Fig. 2c).

The synchrotron-based X-ray total scattering measurements and pair distribution function (PDF) analysis are powerful methods to characterize disordered or amorphous structures[35]. The as-synthesized SnTi–$Fe_2O_3$ sample shows a peak shift, along with the broadening of the peaks, towards a larger interatomic distance ($r$) compared to the undoped $Fe_2O_3$ (Fig. 3d and Supplementary Fig. 11), suggesting an expansion of the hematite lattice by the replacement of $Fe^{3+}$ sites with larger-sized $Sn^{2+}$ and $Ti^{4+}$ ions. The thermal treatment leads to a lattice rearrangement in the bulk as indicated by the peak shifts toward the $r$ values of the undoped $Fe_2O_3$. Considering that the length of corner-sharing Ti–Ti (and Sn–Sn) bond of rutile $TiO_2$ (and $SnO_2$) is 3.55 Å[36] (and 3.68 Å[37]) along with the corresponding PDFs (Supplementary Fig. 12), an increase in amplitude at 3.5–3.7 Å implies the formation of the dopant oxide overlayers. A new peak with a $Q$ value of 1.872 Å$^{-1}$ (1.916 Å$^{-1}$) for the annealed Sn–$Fe_2O_3$ (Ti–$Fe_2O_3$) is assigned to $SnO_2$ (rutile $TiO_2$) (Supplementary Fig. 12)[37,38]. Meanwhile, only a weak and broad scattering peak located between that of $SnO_2$ and $TiO_2$ was detected for the annealed SnTi–$Fe_2O_3$. This result suggests the formation of SnTiO$_x$ phases, not the simple mixture state of $TiO_2$ and $SnO_2$.

Based on these results, we propose dopant segregation, as illustrated in Fig. 3e. Owing to the sintering of nanocrystal

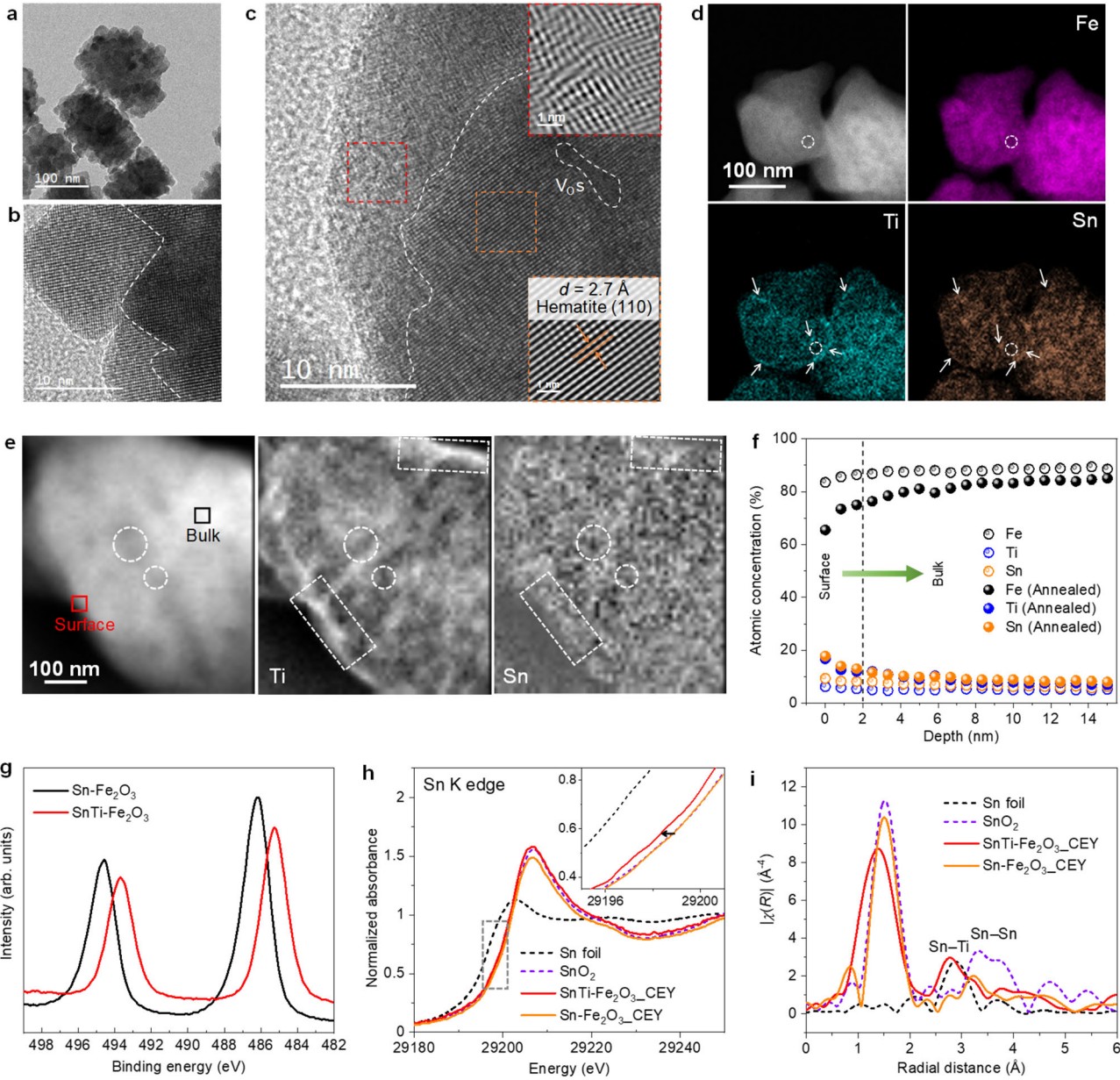

**Fig. 2 Characteristics of hematite MC-derived heterostructures. a** TEM and **b** HRTEM images of as-synthesized SnTi–Fe$_2$O$_3$ MCs. **c** HRTEM image of annealed SnTi-Fe$_2$O$_3$ MC. Insets indicate inverse fast Fourier transform (FT) images of the selected regions with the same colors as indicated by the dashed frames. **d** HAADF-STEM images and corresponding EDX chemical composition maps of annealed SnTi–Fe$_2$O$_3$ MCs. **e** HAADF-STEM image (left) and corresponding EELS composition maps of Ti (451.7–469.7 eV) (middle) and Sn (507.5–525.5 eV) (right) signals of a typical annealed SnTi–Fe$_2$O$_3$ MC. **f** XPS depth analysis of the as-synthesized and annealed SnTi–Fe$_2$O$_3$ samples. **g** Sn 3$d$ XPS spectra of annealed SnTi–Fe$_2$O$_3$ and Sn–Fe$_2$O$_3$ MCs. **h** Ex situ Sn K-edge XANES spectra of the annealed Sn-containing samples measured in CEY mode. **i** The corresponding Sn K-edge FT-EXAFS spectra of the samples.

interfaces (GB elimination) by annealing, interfacial $V_O$ with a positive charge(s) (i.e., $V_O^{\bullet\bullet}$ in the Kröger–Vink notation) are formed, which can create the space charge regions (Supplementary Fig. 13 and Supplementary Table 1) and yield a repulsive force against the dopant cations. This effectively drives the oriented self-segregation of larger-sized dopants to the external surface in addition to the elastic energy (EE) induced by the size mismatch between the dopant and host ions without any significant dopant accumulation in the GB regions (see Supplementary Note 1 for details).

**Formation of composite oxide overlayers**. To elucidate the local structures of the overlayers, X-ray absorption spectroscopy (XAS) at Fe, Ti, and Sn K-edges was performed. The normalized Fe

K-edge XANES spectra (Supplementary Fig. 14) of all the hematite-based samples are close to the standard α-Fe$_2$O$_3$, indicating that the doping and annealing treatments have negligibly affected the valence and coordination states of the iron cation. The normalized Ti K-edge XANES spectra of the Ti-containing samples are shown in Fig. 4a. For transition metals in oxides, the threshold energy position of the spectra is very sensitive to their oxidation states, while the shapes of the peaks give information about the local structural environments of the absorbing elements[39]. The peak shapes of the annealed Ti–Fe$_2$O$_3$ MCs in pre- and post-edge regions are similar to those of rutile TiO$_2$, but the line position is located between those of rutile and FeTiO$_3$. These results are consistent with the fact that the rutile overlayer with the FeTiO$_x$ intermediate layer is seen in the HRTEM image

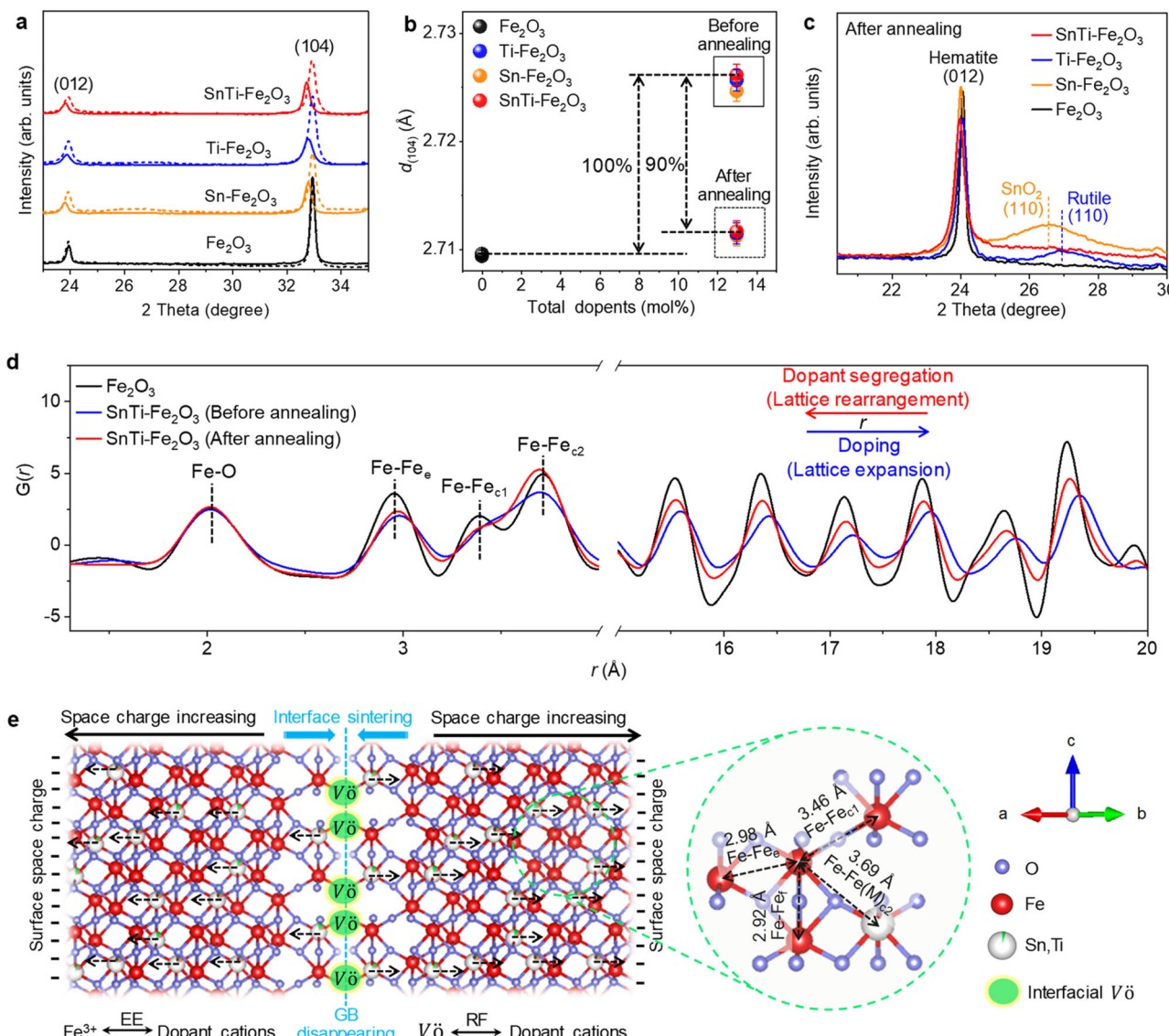

**Fig. 3 Crystallographic analyses of dopant segregation. a** Powder XRD patterns of the samples measured for as-synthesized (solid lines) and annealed (dashed lines) samples with a scanning rate of $10°\ min^{-1}$. **b** Lattice spacing $d_{104}$ values of the samples before and after annealing. **c** Powder XRD patterns of annealed samples measured with a scanning rate of $1.0°\ min^{-1}$. **d** PDF analyses of the samples. The peak at approximately 2 Å is composed of shorter Fe-O (1.94 Å) and longer Fe-O (2.12 Å) distances. The peak at approximately 3 Å is composed of the first neighbor edge-sharing (Fe-Fe$_e$, 2.95 Å) and face-sharing (Fe-Fe$_f$, 2.90 Å) Fe-Fe distances. The first and second neighbor corner-sharing Fe-Fe pairs are at 3.39 Å (Fe-Fe$_{c1}$) and 3.72 Å (Fe-Fe$_{c2}$), respectively[62]. **e** Schematic illustration of the driving force for oriented dopant segregation in hematite during thermal treatment. The self-segregation is driven by the space charges induced by the dopants and $V_O$, RF between the dopant cations and positively charged $V_O$, and EE induced by the size mismatch between the dopant and the host cations.

(Supplementary Fig. 7). For the annealed SnTi–$Fe_2O_3$ sample, the characteristic peak of rutile in the pre-edge (indicated by the black arrow) did not appear. Moreover, the peak shape and position are different from those of reference $TiO_2$ and annealed Ti–$Fe_2O_3$, indicating that no rutile $TiO_2$ phase was formed as the main product in this sample. The annealed SnTi–$Fe_2O_3$ sample also has a similar strongest absorption peak with $FeTiO_3$ at the post-edge region; thus, the oxidation state of Ti ions is similar to that of $FeTiO_3$. Besides, a much stronger first post-edge peak may indicate the Sn–Ti coordination. This was further supported by the comparison of the ex situ Ti K-edge XAS profile measured in transmission and CEY modes (Supplementary Fig. 15).

Figure 4b depicts in situ Ti K-edge XANES spectra of as-synthesized SnTi–$Fe_2O_3$ MCs. When the temperature increased from 40 to 700 °C, a negligible change in the peak shape and

position was observed. No characteristic peak of the rutile phase was observed during heating, whereas the two post-edge peaks shifted to the higher energy and became smooth with an increase in the first post-edge peak at 700 °C due to the formation of binary Sn–Ti oxides. For the Ti–$Fe_2O_3$ sample, the rutile phase was formed at 700 °C (Supplementary Fig. 16).

Ex and in situ Sn K-edge FT-EXAFS measurements enhanced the investigation of the dynamics of dopant segregation (Figs. 2i and 4c, and Supplementary Figs. 17 and 18). The peak position of the Sn–Sn bond for annealed Sn–$Fe_2O_3$ sample is close to that of the reference $SnO_2$ (Fig. 2i and Supplementary Fig. 17), proving the formation of $SnO_2$ overlayers. For the annealed SnTi–$Fe_2O_3$ sample, the second main shell has a shorter radial distance than that of $SnO_2$ owing to the formation of Sn–Ti coordination (i.e., $SnTiO_x$)[30,31]. The peak intensity of the Sn–Fe coordination shell

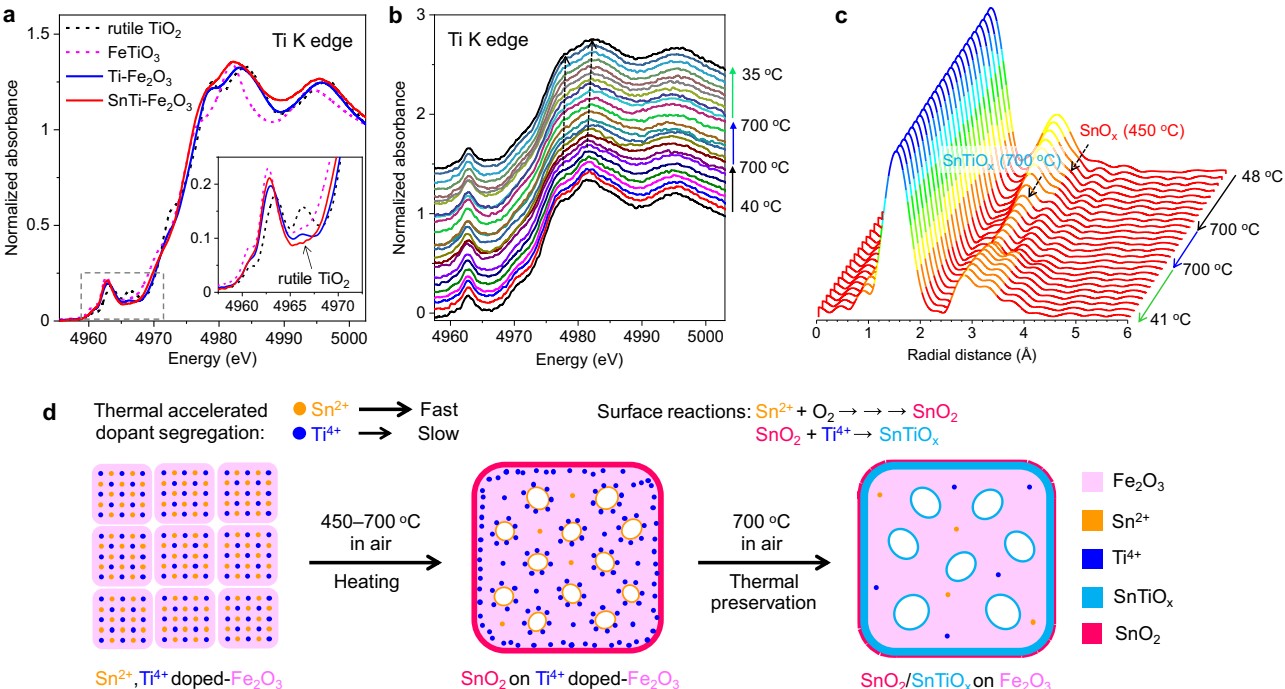

**Fig. 4 In situ observation of dopant segregation. a** Ex situ Ti K-edge XANES spectra of annealed Ti-containing samples and reference samples. **b** In situ Ti K-edge XANES spectra of as-synthesized SnTi–Fe$_2$O$_3$ measured with the similar heating procedure used for electrodes preparation. The spectra are shifted along the y-axis for the sake of better clarity. **c** In situ FT-EXAFS spectra of as-synthesized SnTi–Fe$_2$O$_3$ measured with the similar heating procedure used for electrodes preparation. **d** Schematic illustration of the binary dopant segregation enabling the heterostructures during thermal treatment of as-synthesized SnTi–Fe$_2$O$_3$ MC.

gradually decreases as the temperature increases from 48 to 700 °C (Fig. 4c) because the Sn-coordination becomes disordered due to the diffusion of Sn ions from the hematite lattice. A second Sn–Sn coordination peak is formed and grows when the temperature rises to ~450 °C (Supplementary Fig. 19). Meanwhile, the Sn–Ti coordination peak appears at 700 °C and becomes stronger during the heat preservation and cooling stages, which agrees with the Ti K-edge XANES result (Fig. 4b). The growth of the Sn–Sn and Sn–Ti coordination peaks during the cooling stage indicates ordered Sn-coordinations by suppressed oscillation or diffusion of the elements at lower temperatures. For Sn–Fe$_2$O$_3$ sample, the SnO$_2$ phase is formed, as indicated by the increase in the Sn–Sn coordination at ~550 °C (Supplementary Fig. 18). Based on these results, it is concluded that Sn$^{2+}$ ions migrate from the hematite lattice before Ti$^{4+}$ ions owing to their larger radius (i.e., larger EE), and segregate on the surface to form the SnO$_2$ (or SnO) phase at ~450 °C. The deficient SnTiO$_x$ overlayers are then formed when Ti$^{4+}$ ions segregate at the surface and react with SnO$_2$ at 700 °C (Fig. 4d). In addition, a very small amount of SnO$_2$ remained at the outer surface during the annealing treatment at 700 °C.

**Performance of PEC H$_2$O$_2$ synthesis**. Next, we demonstrate an outstanding ability of the hematite-based heterostructure as a photocatalyst for PEC H$_2$O$_2$ synthesis (Fig. 5a). Figure 5b shows the current density–voltage curves of photoanodes, which were prepared by spin-coating a suspension of as-synthesized MCs followed by the same thermal treatment as mentioned earlier, operated in the dark or under simulated sunlight illumination. The photocurrent density obtained for the optimized SnTi–Fe$_2$O$_3$ photoanode at 1.23 V vs. RHE was approximately 1.1 mA cm$^{-2}$,

Ti–Fe$_2$O$_3$ (0.83 mA cm$^{-2}$), and Sn–Fe$_2$O$_3$ (0.23 mA cm$^{-2}$). This improved performance may result from the reduced electron transfer resistances in the bulk and at the interfaces of hematite/ FTO and hematite/electrolyte by the surface passivation with overlayers and n-type conductivity arising from the interfacial V$_O$ in addition to interparticle sintering and necking[14,15], as suggested by the electrochemical impedance spectroscopy (EIS) (Supplementary Fig. 20) and Mott–Schottky plots (Supplementary Fig. 13 and Supplementary Table 1).

We quantitatively analyzed the H$_2$O$_2$ generation by the N,N-diethyl-1,4-phenylenediamine (DPD) method (Supplementary Fig. 21)[40]. The number of H$_2$O$_2$ produced from the SnTi-Fe$_2$O$_3$ photoanode increased almost linearly with the illumination time, which is much more active than Ti–Fe$_2$O$_3$, Sn–Fe$_2$O$_3$, and Fe$_2$O$_3$ (Fig. 5c). No H$_2$O$_2$ was generated without light irradiation or from the Pt cathode under the operating conditions (Supplementary Fig. 22a). The Sn–Fe$_2$O$_3$ photoanode (i.e., the SnO$_2$ overlayer) showed a much lower photocurrent but a higher Faradaic efficiency (FE) of H$_2$O$_2$, while the Ti–Fe$_2$O$_3$ photoanode (i.e., the TiO$_2$ overlayer) exhibited an opposite trend (Fig. 5d). No gaseous oxygen was detected by the gas chromatography (GC) analysis, possibly due to the low concentration of evolved O$_2$, whereas the amount of H$_2$ (with FE over 90%) linearly increased with the illumination time (Supplementary Fig. 22b, c).

Furthermore, the co-doping of Sn and Ti significantly enhanced the generation of both photocurrent (i.e., H$_2$) without any notable decrease over 90 min and H$_2$O$_2$ with high FEs (>90%) in the range of 1.0–1.8 V vs. RHE (Supplementary Fig. 23). The optimized SnTi–Fe$_2$O$_3$ photoanode realized an H$_2$O$_2$ generation rate of ~0.8 μmol min$^{-1}$ cm$^{-2}$, which is comparable to those of the active BiVO$_4$-based photoanodes (Supplementary Fig. 24). To the best of our knowledge, this is the first example of a hematite-based PEC water-splitting system that

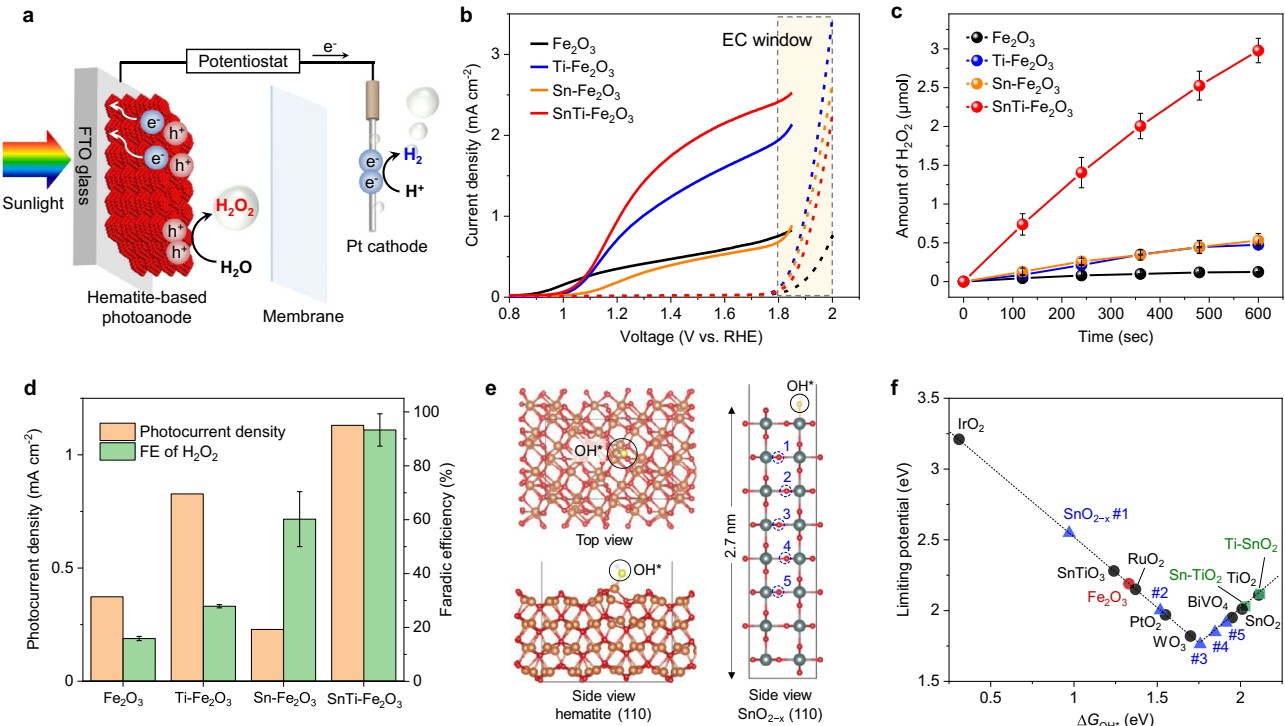

**Fig. 5 PEC H₂O₂ synthesis and DFT calculations. a** Illustration of PEC water splitting system using a hematite-based photoanode. **b** Current density–voltage curves of Fe₂O₃, Sn–Fe₂O₃, Ti–Fe₂O₃, and SnTi–Fe₂O₃ photoanodes in 1.0 M NaHCO₃ under dark and back illumination with AM 1.5 G simulated sunlight. **c** Amounts of H₂O₂ generated from the photoanodes at 1.23 V vs. RHE with an increase in illumination time. The error bars represent the standard deviation. **d** The photocurrent densities and the FEs of H₂O₂ obtained by different electrodes with back illumination at 1.23 V vs. RHE. The error bars represent the standard deviation. **e** Structural models of hematite (110) (left) and SnO₂₋ₓ (110) (right) for DFT calculations. **f** Activity volcano plots based on calculated limiting potentials as a function of $\Delta G_{OH^*}$. The calculated values are summarized in Supplementary Table 3.

achieves a high FE for H₂O₂ generation with a good H₂/H₂O₂ co-production capability. At higher voltages (>1.8 V vs. RHE), dark currents are more significant owing to the EC oxidation of water for obtaining O₂ (Fig. 5b)[20], leading to the decreased FE of H₂O₂ down to ~65% (Supplementary Fig. 23c). Here, we notice that SnTi–Fe₂O₃ photoanode exhibits the lowest activity for EC oxidation among the doped samples, implying that the hetero-overlayer has a specific structure suitable for H₂O₂ generation.

**DFT calculations of adsorption energies.** To identify reaction active sites, we calculated the adsorption energies of relevant intermediates O* and OH* for various structures, including pure hematite (Fig. 5e), using density functional theory (DFT) based on the computational hydrogen electrode model[41,42]. We then calculated free energy changes of OH* and O* ($\Delta G_{OH^*}$ and $\Delta G_{O^*}$) to construct activity volcano plots for two-electron water oxidation to H₂O₂.

As shown in Fig. 5f, bare hematite (110) surface is not suitable for H₂O₂ evolution, but it is for O₂ evolution (Supplementary Figs. 25 and 26, Supplementary Table 2, and Supplementary Note 3), judging from the calculated $\Delta G_{OH^*}$ (1.33 eV) and $\Delta G_{O^*}$ (3.45 eV), which are deviated from the preferred range for H₂O₂ evolution (1.6 eV < $\Delta G_{OH^*}$ < 2.4 eV, $\Delta G_{O^*}$ < 3.5 eV)[19,43]. We also point out that a simple doping treatment cannot improve the reaction selectivity, according to the fact that the $\Delta G_{OH^*}$ values calculated for two local structures, Sn⁴⁺-doped TiO₂ (Sn-TiO₂) and Ti⁴⁺-doped SnO₂ (Ti-SnO₂), where the dopants are considered as the surface active sites, are comparable to those of SnO₂ and TiO₂, respectively (Fig. 5f, Supplementary Fig. 27, and Supplementary Table 3).

Recently, Diehl et al. reported an ilmenite-type SnTiO₃ structure where each Sn²⁺ possesses a lone pair, forming layers separated by a

van der Waals gap[44,45]. This finding inspired us to explore local structures of SnTiO_x overlayers, but our calculations revealed that the ideal SnTiO₃ (0001) surface has a $\Delta G_{OH^*}$ of 1.24 eV ($\Delta G_{O^*}$ of 3.04 eV), which is not suitable for H₂O₂ evolution (Fig. 5f, Supplementary Fig. 28, and Supplementary Table 3). They also reported that an oxidized passivation layer (~2 nm thickness) that resembles SnO₂ formed at the top surface of SnTiO₃. We thus modeled various rutile SnO₂ structures possessing V_O, where subsurface V_O mimics the Sn²⁺ support from SnTiO₃ and examined their possibility as a catalytic site on the annealed SnTi-Fe₂O₃. As demonstrated in Fig. 5e and f, when V_O was introduced near the surface of SnO₂ (site #1), the OH adsorption was significantly enhanced, leading to a poor H₂O₂ evolution activity. The two electrons are left behind when a neutral V_O is formed and these electrons are delocalized over neighboring Sn⁴⁺ [46,47]. An increase in electron density on the Sn ions alters their coordination from 6- to 5-fold coordination. This local under-coordination increases the electronegativity of Sn and reduces the reorganization energy required to distort Sn–O bonds for OH adsorption. Meanwhile, when V_O is present in deeper positions, the $\Delta G_{OH^*}$ value shifts toward the volcano peak where the catalyst is optimal for H₂O₂ production according to Sabatier's principle (Fig. 5e, f, Supplementary Fig. 29, and Supplementary Table 4)[48]. We also found a similar tendency for rutile-type Sn₀.₅Ti₀.₅O₂₋ₓ with V_O (Supplementary Fig. 30 and Supplementary Table 5)[30].

## Discussion

The activity of heterogeneous photocatalysts is strongly influenced by their ability to exhibit chemisorption on reactants and intermediates. Since the surface Sn²⁺ ions are probably oxidized to Sn⁴⁺ during the annealing treatment in the air (Supplementary

Fig. 8d), partially amorphized $SnO_2$ (or $Sn_{0.5}Ti_{0.5}O_2$) (below 2 nm thickness) could form at the outer surface of disordered $SnTiO_x$ overlayers. Such a heterostructure could be realized by successive binary dopant segregation through nanoparticle networks in the MCs (Fig. 4d). Among the structures utilized in the DFT calculations, the prospective ones are $SnO_{2-x}$ or $Sn_{0.5}Ti_{0.5}O_{2-x}$ with $V_O$ at depths of 1.2–1.7 nm (e.g., site #5 in Fig. 5e, f, and Supplementary Fig. 29), which are structurally analogous to the $Sn^{4+}$ species on the disordered $SnTiO_x$ overlayers. Our calculations further point out the importance of the $V_O$ location in controlling the selectivity of the water oxidation reaction (Fig. 5f). It was reported that surface $V_O$ can lower the $H_2O_2$ evolution activity by promoting water dissociation to form intermediates for $O_2$ evolution on $BiVO_4$[49], but the $V_O$-position dependence of $\Delta G_{OH*}$ has been overlooked so far.

For PEC $H_2O_2$ synthesis, several mechanisms have been proposed (Supplementary Note 4)[50]. In particular, the presence of $HCO_3^-$ is significant to accelerate the water oxidation to $H_2O_2$. Baek et al. also demonstrated that the tuning of $\Delta G_{OH*}$ by doping of Gd ions to $BiVO_4$ significantly improves the activity and selectivity of PEC $H_2O_2$ synthesis in 2 M $KHCO_3$ electrolyte. In our system, the oxidized layers supported on $SnTiO_x$ serve as a bifunctional catalyst to adequately adsorb water molecules (i.e., OH*) and facilitate $HCO_3^-$-mediated interfacial transfer[51] of photo-holes from excited hematite core for efficient $H_2O_2$ evolution. In the future, detailed analyses of reaction intermediates using operando spectroscopic methods will be important to refine the mechanism of $H_2O_2$ formation.

In summary, we developed MC-based binary dopant segregation to construct heterostructures with preferential properties for solar $H_2O_2$ synthesis. The oxidized surface $Sn^{4+}$ species on the disordered $SnTiO_x$ overlayers are potential active sites for efficient $H_2O_2$ generation. The composite overlayers on the hematite can be modified to further improve the PEC performance for practical use and fit specific other sustainable reactions like $CO_2$ reduction[52]. Moreover, other types of overlayers, such as nitrides and hydrides[53,54], may be fabricated by varying the synthesis conditions (e.g., annealing in $N_2$ or $H_2$ atmosphere) for emerging functionalities.

## Methods

**Synthesis of hematite-based MCs.** The hematite MCs containing $Sn^{2+}$ (6.5 mol %) and $Ti^{4+}$ (6.5 mol%) dopants ($SnTi–Fe_2O_3$) were synthesized via a simple surfactant-free solvothermal method (Supplementary Fig. 3)[14]. Briefly, a mixed metal precursor of 1 mmol of $Fe(NO_3)_3 \cdot 9H_2O$ (FUJIFILM Wako Pure Chemical, 99.9%), 0.075 mmol of $TiF_4$ (Sigma-Aldrich), and 0.075 mmol of $SnCl_2$ (FUJIFILM Wako Pure Chemical, 97.0 + %) were dissolved in a mixed solvent of 40 mL of $N,N$-dimethylformamide (FUJIFILM Wako Pure Chemical, 99.0 + %) and 10 mL of methanol (FUJIFILM Wako Pure Chemical, 99.8 + %). The above solution was then treated at 180 °C for 24 h in a 100 mL Teflon-lined autoclave reactor. After naturally cooling, the resulting solid product was thoroughly washed with water and ethanol and dried at 60 °C (8 h). The hematite MCs with individual dopants of $Sn^{2+}$ ($Sn–Fe_2O_3$) and $Ti^{4+}$ ($Ti–Fe_2O_3$), and undoped hematite ($Fe_2O_3$) MCs were synthesized via the same method by varying the dopant precursors. The doping levels of $Sn^{2+}$ and $Ti^{4+}$ were controlled by varying the amounts of $Sn^{2+}$ and $Ti^{4+}$ in the precursor solution.

**Fabrication of hematite MC-based photoanodes.** The hematite MC-based films were prepared by multiple spin-coating (3000 rpm) of an ethanol solution containing the highly dispersed hematite-based MCs (10 mg mL$^{-1}$) on a piece of cleaned fluorine-doped tin oxide (FTO)-coated glass (2.5 × 1.7 cm). To obtain the stable films, the above-prepared electrodes were annealed in a furnace at 700 °C for 20 min in the air with a heating rate of 20 °C min$^{-1}$ and collected for further use after naturally cooling. This annealing treatment is also a key step for the formation of dopant oxide overlayers by accelerating the dopant segregation to the hematite surface.

**Characterizations.** Atomic force microscopy (AFM) and Kelvin probe force microscopy (KPFM) measurements were performed on a Dimension Icon (Bruker) using a silicon nitride probe (Bruker, PFQNE-AL). FE-SEM observations were performed on JSM-7100F (JEOL). TEM observations were performed on JEM-2100F (JEOL) operated at 200 kV. EDX mapping and HAADF-STEM images combined with EELS analysis were collected on a JEM-ARM200F Cold FEG (JEOL) microscope operated at 200 kV. The powder XRD patterns were recorded on a Rigaku Ultima IV diffractometer with Cu Kα radiation ($\lambda = 1.5418$ Å) at a voltage of 40 kV and a current of 40 mA. The XPS measurements were performed on a PHI X-tool (ULVAC-PHI). The spectra were calibrated by the reference of the C1s peak at 284.8 eV. Synchrotron-based X-ray total scattering measurements with PDF analysis were performed with the incident X-ray energy of 61.4 keV at BL04B2 beamline in SPring-8, Japan. The data were collected using the hybrid detectors of Ge and CdTe. The reduced PDF $G(r)$ was obtained by the conventional Fourier transform of the Faber–Ziman structure factor $S(Q)$[55] extracted from the collected date[56]. Ex and in situ X-ray absorption spectral measurements were performed at BL01B1 beamline in SPring-8, Japan. The samples were prepared by pelletizing the uniform mixtures of hematite-based MCs or reference samples with dehydrated boron nitride powders. The collected data were processed using the IFEFFIT software package. The filtered $k^3$ weighted $\chi$ spectra were Fourier transformed into $r$ space ($k$ range: 3.0–16.0 Å$^{-1}$ for Fe K, 1.0–6.8 Å$^{-1}$ for Ti K, and 1.0–9.0 Å$^{-1}$ for Sn K).

**PEC measurements.** The PEC measurements were conducted in a typical three-electrode system in 1.0 M $NaHCO_3$ aqueous solution (pH = 8.3) with the fabricated hematite MC-based film as working electrode, Pt wire as the counter electrode, and Ag/AgCl electrode with saturated KCl solution as the reference electrode. The reactors with photoanode and cathode were separated by a membrane film (Sigma-Aldrich, Nafion® 117). All the electrochemical data were recorded and analyzed on an electrochemical workstation (ALS, model 608E). A xenon light source (Asahi Spectra, LAX-C100) equipped with an AM 1.5 filter with the light intensity of 100 mW cm$^{-2}$ (calibrated by a silicon photodiode detector (Asahi Spectra, CS-30)) was used to irradiate the photoanodes from the backside (through the FTO glass) with a working area of 0.72 cm$^2$. The current–voltage curves were obtained by cyclic voltammetry at a scan rate of 20 mV s$^{-1}$. All the applied potentials have been converted into the potential vs. RHE via the Nernst equation ($E_{RHE} = E_{Ag/AgCl} + 0.059 \times pH + E°_{Ag/AgCl}$). The EIS measurements were performed at 1.23 V vs. RHE with frequencies between 0.5 Hz and 10 kHz under back illumination with the simulated sunlight. The Mott–Schottky plots were measured in the dark at a frequency of 10 kHz.

**Product analysis.** The evolved gas from the PEC cell was analyzed by a GC (Shimadzu, GC-8A) equipped with an MS-5A column and a thermal conductivity detector. To remove air in the reactor, the electrolyte solutions in well-sealed working and counter electrode cells that were separated by the Nafion membrane were bubbled with Ar gas for at least 30 min before the measurements. To minimize the influence of air leaks, the analysis of gas products was carried out independently using the same electrode under the same conditions. A relatively large amount of gas sample (50 μL) from the working electrode cell was measured every time for $O_2$ detection. 20 μL of gas sample from the counter electrode cell was collected and measured for $H_2$ analysis. The GC was calibrated by injecting exact volumes of pure $O_2$ and $H_2$ gases (Supplementary Fig. 32). $H_2O_2$ evolution was evaluated using $N,N$-diethyl-1,4-phenylenediamine (DPD) sulfate. Typically, 200 μL sample aliquots collected with a syringe during irradiation were mixed with 200 μL potassium phosphate buffer solution (pH = 7), 2 mL water, 20 μL DPD solution (0.1 g DPD in a 10 mL 0.05 M $H_2SO_4$ solution), and 20 μL freshly prepared peroxidase (POD) solution (3 mg POD in 3 mL deionized water), and the mixtures were shaken for 120 s. The above solutions were analyzed by UV-vis spectroscopy (JASCO, V-700). The $H_2O_2$ concentrations were calibrated with standard $H_2O_2$ solutions. The concentration of dissolved $O_2$ in the electrolyte was evaluated using a free radical analyzer (World Precision Instruments, TBR4100), which is based on an electrochemical (amperometric) detection, with an ISO-OXY-2 oxygen sensor (World Precision Instruments). The detected currents were converted to $O_2$ concentrations using the obtained calibration curve.

**DFT calculations.** All the calculations reported in this work were carried out using the revised version of Perdew-Burke-Ernzerhof exchange-correlation functional[57] and the projector-augmented wave method as implemented in Vienna Ab Initio Simulation Package[58,59]. The spin-polarization effect and dipole correction were both taken into account. Convergence was considered to be achieved when the energy difference was less than $10^{-6}$ eV and the force was less than $10^{-2}$ eV per Å for geometry optimization. The (110) slab model of $Fe_2O_3$ was prepared by first performing the geometry optimization of the anti-ferromagnetic bulk system with a cut-off energy of 520 eV and Monkhorst–Pack k-point meshes of 4 × 4 × 4 to determine the lattice constants, and then by using 4 Fe-layers with a vacuum of 15 Å and 4 × 4 × 1 Monkhorst–Pack k-points. For the calculations of OH* and O*, several coverage patterns have been considered, from 1/12 to 0.5, corresponding to 1 and 6 adsorbates per cell. For $Fe_2O_3$, a Hubbard $U$ value of 4.3 eV was employed[60]. The computational hydrogen electrode was employed, for which the Gibbs free energy of $1/2 H_2 \rightleftharpoons H^+ + e^-$ is set to zero. The free energy change was computed as $\Delta G = \Delta E + \Delta ZPE - T\Delta S$, where previously reported values are used for the zero-point energy and entropy to compute $\Delta ZPE - T\Delta S$ at standard

conditions[61], since they are known to be very similar between different oxides. For the calculations of $SnO_{2-x}$, the same procedure as $Fe_2O_3$ was employed, except that the cut-off energy used was 400 eV and the (110) slab model consisted of $1 \times 2$ super-cell with 8 Sn-layers. The adsorption energies were computed with one adsorbate for each cell. The $V_O$ dependence of the volcano plot was investigated with changing vacancy sites (Supplementary Fig. 28).

## Data availability

Source data are provided with this paper.

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

## Acknowledgements

Mr. Hideo Maruyama (KANEKA TECHNO RESEARCH CORPORATION) is acknowledged for AFM/KPFM measurements. The X-ray total scattering and absorption measurements were carried out at SPring-8 under the proposals 2020A1208 and 2021A1114, and 2020A1209 and 2021A1113, respectively. This work was partially supported by Nagoya University microstructural characterization platform as a program of "Nanotechnology Platform" of the Ministry of Education, Culture, Sports, Science and Technology (MEXT), Japan, MEXT as "Program for Promoting Researches on the Supercomputer Fugaku" (Realization of innovative light energy conversion materials, Grant Number JPMXP1020210317), JST A-STEP Grant Number JPMJTR20TD, JSPS KAKENHI Grant Numbers JP18H01944, JP20H04673, JP21H02049, and others. KANEKA CORPORATION is acknowledged for financial support. Zhujun Zhang thanks to the financial support from Marubun Research Promotion Foundation.

## Author contributions

Z.Z. and T. Tachikawa wrote the paper with contributions from all co-authors. Z.Z. synthesized the materials and performed most of the experiments. Z.Z. and T. Tachikawa performed the data analysis. T. Tsuchimochi and S.L.T. performed the DFT calculations. T.I. performed the X-ray absorption measurements. Y.K. performed parts of the PEC experiments and data analysis. S.M. performed the STEM-EELS measurements. K.O. and H.Y. performed the X-ray total scattering measurements. T. Tachikawa conceived and supervised the project.

## Competing interests

The authors declare no competing interests.
