## [Peer Review File · Nature Communications]

REVIEWER COMMENTS

Reviewer #1 (Remarks to the Author):

The authors here present an article on the synthesis of doped hematite and their application for photoelectrochemical H₂O₂ formation. They do an excellent job in the synthesis and physicochemical characterisation of the material, but the article is weak on the photoelectrochemical part which is the most novel part. Please see my following comments:

1. Attention should be paid to the fact the experiments are done in a 3-electrode system. You don't strictly split water in a 3-electrode system. The working electrode is connected to the potentiostat and the electrons moving to the potentiostat are not the same electrons reaching the counter electrode. If you switch off the potentiostat, you won't see any products... The drawing in Fig 5a saying "external bias" would be the case if you for example connect a solar cell in series, but in this article they are connecting a potentiostat... The potentiostat pushes electrons to the counterelectrode and will apply any needed voltage to make sure this happens... This is very different from a discrete "external bias" a solar cell would provide. Therefore, please update Fig 5a and carefully report the H₂ production. It is irrelevant what happens in the counterelectrode in the presence of a potentiostat...(but see my comment later on FE). It is only relevant if you put a solar cell in series or if you have a 2 electrode system.

2. The O₂ is said to be measured by GC TCD. How is this experiment carried out reliably and with no interference of room air? Nothing is explained on this in the paper and it is known O₂ measurement by GC at those levels is quite complicated, so often fluorescence O₂ sensors are used instead. And a TCD is not a precise detector, unlike others such as BID. And authors are taking samples for H₂O₂ measurement at the same time that can introduce air leaks...

3. This high FE value on Fe₂O₃ is rarely reported. The author should provide more information, such as the ΔG_{OH^*} (critical for the selectivity of H₂O₂).

4. The author should mention if they separate the counter electrode and working electrode by a membrane in the 3-electrode cell. I assume they don't, then they should use the H₂ generation amount to calculate the FE again. Because H₂O₂ can be produced through ORR as well. So if the working electrode generates some O₂, it is possible to produce H₂O₂ on the counter electrode by using the generated O₂ and water. This will give a wrong calculation of the FE. Please refer to articles where they do this correction.

5. Figure 1: the 'electrical potential' is not what they mean. Please check the paper: ACS Energy Lett. 2021, 6, 261–266, “Potentially Confusing: Potentials in Electrochemistry”

6. Oxygen vacancies on the surface of hematite are recombination centres. They are harmful to interfacial(electrolyte/semiconductor) charge transfer. I agree with one of your claim that a high doping level at the interface between crystal grains helps charge migration. However, a high doping level on the surface narrows the space charge layer and creates a lot of oxygen vacancies. Both will increase the surface recombination. This worse charge injection can explain the anodic shift onset potentials observed from the photocurrent, but this is inconsistent with the smaller semicircles in the PEIS plot. It is known that intentional Sn doping of hematite is not very successful, and better results have been achieved by letting Sn from the FTO/ITO diffuse at 800 degC. The authors should comment on these aspects.

7. BiVO₄ is said to be expensive, but this is arguable

8. Please provide details about the ramp rate to heat and cool the photoanodes

Reviewer #2 (Remarks to the Author):

This manuscript reports a hematite-based heterostructure photoanode that oxidizes water to H₂O₂ with nearly 100% faraday efficiency. The manuscript is well written and the research was carefully done with a thorough set of experiments.

1. The several materials compared will have different bandgaps and band positions, so the light may affect their charge transfer efficiency differently. The authors should measure the faraday efficiencies of those materials under dark as well.

2. The manuscript mentions that SnTiO_x has a high faraday efficiency because its oxygen vacancy is the preferred adsorption site for HCO₃⁻. The authors need to explain this in detail with some supporting evidence. Also, those oxygen vacancies may not be stable under high anodic conditions used to produce H₂O₂. A longer stability test needs to be conducted to see how faraday efficiency changes with time.

Reviewer #3 (Remarks to the Author):

In this work the authors reported the preparation of heterostructures consisting of photoactive core and co-catalyst shell for efficient photoelectrochemical H₂O₂ production by using thermal annealing of metal-doped hematite mesocrystals. The catalyst has been carefully characterized by using various techniques. However, similar catalysts and the preparation methods have been previously reported in their own work and in this work it is just a new application for the production of H₂O₂. Thus the working mechanism should be clearly clarified to understand that why the present catalyst can be efficiently used for the H₂O₂ production, while the typical hematite shows a low performance. Actually, hematite modified by Ti and Sn has been widely reported for PEC water splitting. Some other critical issues should also be carefully addressed:

1. The EELS spectra in Fig. 2f and 2g are not good enough and the important features are not clear. Since the main difference comes from the surface signal, the reviewer will suggest to perform the soft X-ray spectra (TEY mode) at Sn and Ti L-edges to clearly identify the features. Actually, the signal is too noisy to make the corresponding conclusions. For example, the O K-edge spectra with a lower first peak are very normal for hematite, which cannot be used to identify the presence of oxygen vacancy. Pure hematite without oxygen vacancy will also show the lower first peak. The energy differences such as 10.5 eV and 7.8 eV are also not clear due to the noisy signal, which cannot be used to make a solid conclusion.
2. The Sn-doping in the samples before annealing might be not true according to the Sn K-edge XAS spectra (Fig. S15). If Ti and Sn are doped in Fe₂O₃, the Ti and Sn atoms in Fe₂O₃ will be in similar environment as that in Fe₂O₃, which means that the second shell Ti-Fe or Sn-Fe peak position and shape will be similar to that of Fe₂O₃ (see Fig. S14). However, Sn in Fig. S15 shows the environment similar to that in SnO₂, suggesting that the formation of SnO₂ in hematite instead of Sn-doping. It could be amorphous SnO₂ clusters in hematite which can also change the crystal structure. Ti K-edge EXAFS data should also be provided to confirm the Ti-doping.
3. The XPS shift of Sn and Ti may come from the wrong alignment of the XPS peak position. The two elements shift to the same energy direction with similar values.
4. It is strange that the authors compared the XRD peak difference around 33 degree for the samples before annealing, while after annealing the XRD peaks are compared at around 24 degree. Both peaks should be compared before and after annealing.

5. Fig. 4a is not clear. So many curves are put together and it is hard to identify the difference. Moreover, the spectra of Ti-Fe₂O₃ and SnTi-Fe₂O₃ are almost identical to that of rutile TiO₂, suggesting the rutile structure. The white line peak position of Ti-Fe₂O₃ is almost the same as that of rutile TiO₂.

6. In Fig. 4a the absence of pre-peak (the arrow) in the spectrum of SnTi-Fe₂O₃ might be related to the formation of amorphous TiO₂ with a small size. The different pre-peak position should be clarified based on the XANES spectral fitting or calculation. It cannot make a direct conclusion for the formation of FeTiO₃. It could also be Fe₂TiO₅. The in-situ spectral changes might be explained by the aggregation of small amorphous TiO₂ to rutile TiO₂.

7. For the Sn K-edge EXAFS spectra, the SnO peak is too weak to confirm the existence of SnO. The formation of SnO or SnTiO_x will also change the first shell peak in the EXAFS spectra, but no clear evidence can be observed. The formation of SnTiO_x should be confirmed by the spectral fitting based on the corresponding models instead of a simple hypothesis. The present spectral changes are not a strong evidence for the formation of SnTiO_x. The Sn-Ti coordination cannot be proved in the whole manuscript.

8. The stable time in Fig. S18a is too short. It should be at least more than 1 hour for a stable catalyst.

Responses to the comments from Reviewer #1:

The authors here present an article on the synthesis of doped hematite and their application for photoelectrochemical H₂O₂ formation. They do an excellent job in the synthesis and physicochemical characterisation of the material, but the article is weak on the photoelectrochemical part which is the most novel part. Please see my following comments:

Reply: We appreciate the reviewer for the high evaluation of our work. To improve the part of the photoelectrochemical (PEC) H₂O₂ generation, we performed DFT calculations (Figs. 5e and f, Supplementary Figs. 24–27). The results demonstrate the significant role of hetero-overlayer in improving PEC H₂O₂ production performance of hematite.

1. Attention should be paid to the fact the experiments are done in a 3-electrode system. You don't strictly split water in a 3-electrode system. The working electrode is connected to the potentiostat and the electrons moving to the potentiostat are not the same electrons reaching the counter electrode. If you switch off the potentiostat, you won't see any products... THE drawing in Fig 5a saying "external bias" would be the case if you for example connect a solar cell in series, but in this article they are connecting a potentiostat... The potentiostat pushes electrons to the counterelectrode and will apply any needed voltage to make sure this happens... This is very different from a discrete "external bias" a solar cell would provide. Therefore, please update Fig 5a and carefully report the H₂ production. It is irrelevant what happens in the counterelectrode in the presence of a potentiostat...(but see my comment later on FE). It is only relevant if you put a solar cell in series or if you have a 2 electrode system.

Reply: We appreciate the reviewer for helpful explanation and suggestion. We updated Fig 5a and replaced the “external bias” as “potentiostat” in the revised manuscript.

2. The O₂ is said to be measured by GC TCD. How is this experiment carried out reliably and with no interference of room air? Nothing is explained on this in the paper and it is known O₂ measurement by GC at those levels is quite complicated, so often fluorescence O₂ sensors are used instead. And a TCD is not a precise detector, unlike others such as BID. And authors are taking samples for H₂O₂ measurement at the same time that can introduce air leaks...

Reply: Thanks for the reviewer's comments and suggestions. We added the following information “To remove air in the reactor, the electrolyte solutions in well-sealed working and counter electrode cells that were separated by the Nafion membrane were bubbled with Ar gas for at least 30 min before the measurements. To minimize the influence of air leaks, the analysis of gas

products was carried out independently using the same electrode under the same conditions. A relatively large amount of gas sample (50 μL) from the working electrode cell was measured every time for O_2 detection. 20 μL of gas sample from the counter electrode cell was collected and measured for H_2 analysis. The GC was calibrated by injecting exact volumes of pure O_2 and H_2 gases (Supplementary Fig. 29).” in the revised manuscript (page 26). In addition, we measured if any O_2 was generated and reacted with water to produce H_2O_2 during H_2O_2 generation in the counter electrode cell. However, H_2O_2 was not detected in the counter electrode cell under the same conditions (Fig. C1), thus confirming the reliability of the O_2 detection result. This result is presented in Supplementary Fig. 22a.

We also evaluated the concentration of dissolved O_2 in the electrolyte under operation conditions using a free radical analyser (World Precision Instruments, TBR4100), which is based on an electrochemical (amperometric) detection (Fig. C2). The detailed experimental procedures are described in the Methods section. The Faradaic efficiency (FE) of O_2 evolution on the SnTi- Fe_2O_3 MC photoanode was approximately 20%. This value would be reasonable when considering the experimental error in the H_2O_2 analysis. This result is given in Supplementary Fig. 22c.

Fig. C1. H_2O_2 generation in the counter electrode cell.

Fig. C2. Electrochemical detection of dissolved O₂ in 1.0 M NaHCO₃ in the photoanode cell under dark and back illumination with AM 1.5 G simulated sunlight at 1.6 V vs. RHE.

3. This high FE value on Fe₂O₃ is rarely reported. The author should provide more information, such as the ΔG_{OH^*} (critical for the selectivity of H₂O₂).

Reply: According to this suggestion, we calculated the adsorption energies of relevant intermediates O* and OH* for various structures including pure hematite using density functional theory (DFT) based on the computational hydrogen electrode model (e.g. Nat. Commun. 8, 701 (2017)). To identify reaction active structures, we obtained free energy changes in OH* and O* (ΔG_{OH^*} and ΔG_{O^*}) and constructed activity volcano plots. Key findings are summarised as follows.

- 1) Bare hematite (110) surface is not suitable for H₂O₂ generation, but it is for O₂ generation in terms of ΔG_{O^*} (1.96–2.58 eV) and ΔG_{OH^*} (0.62–1.36 eV), regardless of surface adsorption sites (terminal and bridge sites) and coverages (1/12 and 0.5 monolayer coverages). The preferred ΔG_{OH^*} range for H₂O₂ generation was reported as 1.6 eV < ΔG_{OH^*} < 2.38 eV.
- 2) The ΔG_{O^*} and ΔG_{OH^*} for Sn⁴⁺-doped TiO₂ and Ti⁴⁺-doped SnO₂, where the dopants are considered as the active sites, are comparable to those of SnO₂ and TiO₂, respectively.
- 3) The oxygen vacancy close to the adsorption site can significantly enhance the OH* adsorption, suggesting that oxygen vacancy is present in a deeper position.
- 4) Clean SnO₂ has ΔG_{OH^*} of 2.01 eV which is not bad, while surface oxygen vacancy provides too low ΔG_{OH^*} (~0.5 eV). An oxygen vacancy-position dependence of ΔG_{OH^*} was examined to mimic the Sn²⁺ layer supported on deficient SnTiO_{3-x}. Among the modelled structures, SnO_{2-x} with oxygen vacancies at depths of ~1.2–1.7 nm, which is close to the thickness (~2

nm) of the amorphous passivation layer reported for ilmenite-type SnTiO₃ (Chem. Mater. 30, 8932 (2018); Chem. Mater. 33, 2824 (2021)), seems promising.

From these calculations and experimental results, we can conclude that the oxidised surface Sn⁴⁺ species (most probably, a thin layer of SnO₂ on the deficient SnTiO₃ support) are possible active sites for efficient H₂O₂ generation. These results are given in Figs. 5e and f, Supplementary Figs. 25–27 and discussed in the main text (page 17).

4. The author should mention if they separate the counter electrode and working electrode by a membrane in the 3-electrode cell. I assume they don't, then they should use the H₂ generation amount to calculate the FE again. Because H₂O₂ can be produced through ORR as well. So if the working electrode generates some O₂, it is possible to produce H₂O₂ on the counter electrode by using the generated O₂ and water. This will give a wrong calculation of the FE. Please refer to articles where they do this correction.

Reply: Thank you very much for the comment and suggestion. We did separate the counter electrode and working electrode by a Nafion membrane. We added this information in the revised manuscript (page 22). According to the reviewer's suggestion, we also detected the H₂O₂ generation from the counter electrode. However, no H₂O₂ was generated in the counter electrode cell during the measured time period (Fig. C1 and Supplementary Fig. 22a).

5. Figure 1: the 'electrical potential' is not what they mean. Please check the paper: ACS Energy Lett. 2021, 6, 261–266, “Potentially Confusing: Potentials in Electrochemistry”

Reply: We appreciate the reviewer for the kind suggestion. We revised the “electrical potential” as “electrostatic potential” in the revised manuscript after carefully reading the highly related references (Nat. Commun. 8, 1417 (2017); J. Appl. Phys. 64, 4516 (1988)).

6. Oxygen vacancies on the surface of hematite are recombination centres. They are harmful to interfacial(electrolyte/semiconductor) charge transfer. I agree with one of your claim that a high doping level at the interface between crystal grains helps charge migration. However, a high doping level on the surface narrows the space charge layer and creates a lot of oxygen vacancies. Both will increase the surface recombination. This worse charge injection can explain the anodic shift onset potentials observed from the photocurrent, but this is inconsistent with the smaller semicircles in the PEIS plot. It is known that intentional Sn doping of hematite is not very

successful, and better results have been achieved by letting Sn from the FTO/ITO diffuse at 800 degC. The authors should comment on these aspects.

Reply: Thanks to the reviewer's kind comments and suggestions. The onset potential of Fe₂O₃ is shifted to cathodic direction as compared to other samples despite showing a much lower photocurrent. This is probably because water oxidation, which is preferential to O₂ (1.23 V vs. RHE), occurred at the hematite surface which requires relatively lower applied voltages than that for water oxidation to H₂O₂ (1.76 V vs. RHE). We agree with the fact that oxygen vacancies on the hematite surface are harmful to charge transfer at the electrolyte/semiconductor. However, in our case, thin hetero-overlayers produced by the annealing treatment can passivate the trap states on the hematite surface. In addition, much higher carrier densities of SnTi-Fe₂O₃ and Ti-Fe₂O₃ induced by Ti-doping and more oxygen vacancies in the bulk are helpful for charge separation in the bulk, thus resulting in the lower charge transfer resistance (Supplementary Fig. 20 (please see smaller semicircles in the EIS plot) and Supplementary Table 1).

We also agree with the reviewer's comment that intentional Sn doping of hematite is not very successful in traditional high-temperature processes due to the much-larger Sn ions than Fe ions. In this work, we proved the successful doping of Sn ions into the hematite lattice via a solvothermal method operated at a relatively low temperature (180 °C). The as-synthesised doped samples showed similar XRD peaks to those of pure hematite as a result of lattice expansion caused by successful doping of larger Sn²⁺ ions; however, XRD patterns of hematite shifted towards smaller diffraction angles. After the annealing treatment, such lattice expansion almost disappeared due to the segregation of Sn dopants to the outer surface to form a SnO₂ overlayer. We proved that the Sn²⁺ ions can more quickly segregate to the hematite surface (~450 °C) compared to Ti⁴⁺. Both X-ray total scattering and X-ray absorption experiments supported these processes. Our results are also consistent with the reviewer's comment that the intentional Sn doping of hematite is not always very successful via a traditional thermal method.

7. BiVO₄ is said to be expensive, but this is arguable.

Reply: We accept the reviewer's comment about BiVO₄. We have revised the relative comments as "The PEC H₂O₂ production has been mostly realised by a two-electron pathway from water oxidation using BiVO₄-based photoanodes;¹⁸⁻²¹ however these photoanodes are still unstable for practical due to the dissolution of V⁵⁺ that arises from anodic photocorrosion.²²" in the revised manuscript (page 4).

8. Please provide details about the ramp rate to heat and cool the photoanodes.

Reply: Thanks to the reviewer's suggestion. In the revised manuscript, we added the details about the ramp rate to heat and cool the photoanodes as follows: "To obtain the stable films, the above-prepared electrodes were annealed in a furnace at 700 °C for 20 min in air with a heating rate of 20 °C min⁻¹ and collected for further use after naturally cooling." in the revised manuscript (page 21).

Responses to the comments from Reviewer #2:

This manuscript reports a hematite-based heterostructure photoanode that oxidizes water to H₂O₂ with nearly 100% faraday efficiency. The manuscript is well written and the research was carefully done with a thorough set of experiments.

Reply: We appreciate the reviewer for the high evaluation of our work.

1. The several materials compared will have different bandgaps and band positions, so the light may affect their charge transfer efficiency differently. The authors should measure the faraday efficiencies of those materials under dark as well.

Reply: We appreciate the reviewer's kind suggestion. We measured the H₂O₂ evolution by the optimised SnTi-Fe₂O₃ photoanode at 1.5 V vs. RHE under a dark environment. However, no H₂O₂ was produced under these conditions (Fig. C3). Therefore, the high selectivity of H₂O₂ generation originates from surface holes generated by light illumination. This result is given in Supplementary Fig. 22a.

Fig. C3. H₂O₂ generation from the SnTi-Fe₂O₃ photoanode under dark at 1.5 V vs. RHE.

2. The manuscript mentions that SnTiO_x has a high faraday efficiency because its oxygen vacancy is the preferred adsorption site for HCO₃⁻. The authors need to explain this in detail with some supporting evidence. Also, those oxygen vacancies may not be stable under high anodic conditions used to produce H₂O₂. A longer stability test needs to be conducted to see how faraday efficiency changes with time.

Reply: Thanks for the suggestion. The mechanism of the H₂O₂ synthesis via water oxidation is still arguable, as we discussed previously. We also noticed that we cannot exclude the possibility that the HCO₃⁻ species on the electrode detected by Raman spectroscopy might partly come from deposited NaHCO₃ salt but not from absorbed HCO₃⁻. Therefore, we removed the Raman data and added the results of DFT calculations to discuss the possible active sites for H₂O₂ generation in the revised manuscript (page 17). In addition, a long-time stability test was added in the revised manuscript (Supplementary Fig. 23d).

Responses to the comments from Reviewer #3:

In this work the authors reported the preparation of heterostructures consisting of photoactive core and co-catalyst shell for efficient photoelectrochemical H₂O₂ production by using thermal annealing of metal-doped hematite mesocrystals. The catalyst has been carefully characterized by using various techniques. However, similar catalysts and the preparation methods have been previously reported in their own work and in this work it is just a new application for the production of H₂O₂. Thus the working mechanism should be clearly clarified to understand that why the present catalyst can be efficiently used for the H₂O₂ production, while the typical hematite shows a low performance. Actually, hematite modified by Ti and Sn has been widely reported for PEC water splitting. Some other critical issues should also be carefully addressed:

Reply: We appreciate the reviewer for the constructive comments and suggestions. In our previous works, we reported the preparation of Ti-Fe₂O₃ via the same method and studied their photoelectrochemical water oxidation to O₂. We agree with the reviewer's comment that we did use similar methods to obtain monometallic oxide overlayer structures. However, the process of dopant segregation in the hematite mesocrystal was not clarified in the previous reports. In this work, we studied the mechanism of dopant segregation in the mesocrystals using various techniques, such as in-situ EXAFS. In addition to the monometallic oxide overlayers of TiO₂ and SnO₂, we developed our strategy to construct novel heterostructures, binary SnTiO_x overlayers

on the hematite mesocrystals, with preferential properties for solar H₂O₂ synthesis. We believe that our study makes a significant contribution to the development of solid photocatalysts because our findings highlight the importance and potential of hierarchical structures with designed hetero-overlayers that are expected to achieve selective solar fuel production. The composite overlayers on the hematite will be modified to further improve the PEC performance for practical use and fit specific other sustainable reactions like CO₂ reduction. Moreover, other types of overlayers, such as nitrides and hydrides, may be fabricated by varying the synthesis conditions (e.g. annealing in N₂ or H₂ atmosphere) for emerging functionalities.

We completely agree with the reviewer's comment that the working mechanism should be clarified to understand the enhancement of H₂O₂ production performance by the present catalyst. Therefore, we performed DFT calculations to identify a possible active structure and clarify the impact of the hetero-overlayers (page 17).

1. The EELS spectra in Fig. 2f and 2g are not good enough and the important features are not clear. Since the main difference comes from the surface signal, the reviewer will suggest to perform the soft X-ray spectra (TEY mode) at Sn and Ti L-edges to clearly identify the features. Actually, the signal is too noisy to make the corresponding conclusions. For example, the O K-edge spectra with a lower first peak are very normal for hematite, which cannot be used to identify the presence of oxygen vacancy. Pure hematite without oxygen vacancy will also show the lower first peak. The energy differences such as 10.5 eV and 7.8 eV are also not clear due to the noisy signal, which cannot be used to make a solid conclusion.

Reply: Thanks to the reviewer's comments and suggestions. We agree with the reviewer's comment that the EELS profiles are not sufficient to provide a solid conclusion. Since the signal from the Sn M_{4,5} edge was very weak compared with that from the O K edge, it is difficult to analyse the chemical state of Sn in detail, although the signal-to-noise ratio of the spectrum has been improved for further analysis. Therefore, we have moved the results of the EELS analysis to the Supplementary Information (Supplementary Fig. 8).

According to the reviewer's suggestion, X-ray absorption measurements were performed using the conversion electron yield (CEY) mode to analyse the surface species (Figs. 2h and i in the revised manuscript, page 8). For the annealed Sn-Fe₂O₃, the pre- and post-edge features are similar to those of the SnO₂ reference, indicating the same oxidation state. In addition, the corresponding Sn-K edge FT-EXAFS spectrum has a second shell that is almost identical to the Sn-Sn shell of SnO₂, indicating the formation of a SnO₂ overlayer. On the other hand, the XAS

spectrum of the annealed SnTi-Fe₂O₃ shows a shift to the lower energy side compared with those of annealed Sn-Fe₂O₃ and SnO₂, indicating the lower valence state of the surface Sn species in the SnTi-Fe₂O₃ sample. In addition, both the first and second shells shifted to lower radial distances (inset of Fig. 2h in the revised manuscript). The Sn K edge EXAFS spectrum of SnTi-Fe₂O₃ shows the main first shell of Sn–O at 1.38 Å and second Sn–Ti shell at 2.76 Å, and weak coordination of Sn–Sn as that of SnO₂ and Sn-Fe₂O₃ at ~3.28 Å (Fig. 2i in the revised manuscript). The above results indicate the formation of deficient SnTiO_x hetero-overlayer with a small amount of oxidised Sn species on the outer surface, as also indicated by the XPS Sn 3d depth profile shown in Supplementary Fig. 8d.

Based on the results, we revised the manuscript as “In addition, the XPS analysis revealed that the valence state of Sn ions for the annealed SnTi-Fe₂O₃ is lower than that (Sn⁴⁺) of the annealed Sn-Fe₂O₃ (Fig. 2g). This was further confirmed by the Sn K-edge X-ray absorption near edge structure (XANES) spectrum of the annealed SnTi-Fe₂O₃ measured in a conversion electron yield (CEY) mode, an effective method to analyse the surface compositions of materials, which indicates a clear shift to lower energy compared with Sn-Fe₂O₃ and SnO₂ (Fig. 2i). The corresponding Sn K-edge extended X-ray absorption fine structure (EXAFS) FT spectrum of SnTi-Fe₂O₃ exhibits the main first shell of Sn–O at 1.38 Å and second Sn–Ti shell at 2.76 Å,^{30,31} in addition to the weak coordination of Sn–Sn as those of SnO₂ and Sn-Fe₂O₃ at ~3.28 Å. The above results indicate the formation of SnTiO_x hetero-overlayer with the possibility of a small amount of SnO₂ at the outer surface, as suggested by the XPS Sn 3d depth profile analysis (Supplementary Fig. 8d).” (page 7) and “However, we noticed that the XPS Sn 3d depth profiles show that the Sn 3d signals slightly shifted to higher binding energies when the sample was etched by Ar for 60–120 s (i.e. 0.83–1.66 nm depth) (Supplementary Fig. 8d), while Ti ions possess the same oxidation state from the surface to the depth of ~7 nm, as indicated by the non-shifted Ti 3d signals (Supplementary Fig. 8c). These results suggest the possibility of existing oxidised Sn ions at the outer surface region (below 2 nm depth). The Sn 3d peaks located at relatively lower energies observed before Ar etching are probably due to the surface adsorbates. Based on the Sn K-edge FT-EXAFS spectra measured in CEY mode (Fig. 2i), such oxidised Sn species might come from a small amount of SnO₂ at the surface.” in the revised Supplementary file (caption of Supplementary Fig. 8, page 11).

Fig. 2. h, Ex situ Sn K-edge XANES spectra of the annealed Sn-containing samples measured in CEY mode. **i**, The corresponding Sn K-edge FT-EXAFS spectra of the samples.

Supplementary Fig. 8. XPS Ti 2p (**c**) and Sn 3d (**d**) depth profiles of the annealed SnTi-Fe₂O₃ measured by Ar etching for different times.

2. The Sn-doping in the samples before annealing might be not true according to the Sn K-edge XAS spectra (Fig. S15). If Ti and Sn are doped in Fe₂O₃, the Ti and Sn atoms in Fe₂O₃ will be in similar environment as that in Fe₂O₃, which means that the second shell Ti-Fe or Sn-Fe peak position and shape will be similar to that of Fe₂O₃ (see Fig. S14). However, Sn in Fig. S15 shows the environment similar to that in SnO₂, suggesting that the formation of SnO₂ in hematite instead of Sn-doping. It could be amorphous SnO₂ clusters in hematite which can also change the crystal structure. Ti K-edge EXAFS data should also be provided to confirm the Ti-doping.

Reply: We agree with the reviewer that the successfully doped Sn and Ti atoms will be in a similar environment to that in Fe₂O₃. The successful doping of Sn and Ti ions into hematite was first proved by XRD results. Both the as-synthesised Sn-Fe₂O₃, Ti-Fe₂O₃, and SnTi-Fe₂O₃ show the

same diffraction peaks as pure hematite with peak shifts to lower diffraction angles (Fig. 3). With the increase in dopant concentration, such a shift becomes larger (Supplementary Fig. 10). This phenomenon can only be explained by doping instead of the formation of their oxide phase. Such peak shifts can be removed after the annealing treatment due to the dopant segregation to the outer surface to form their oxide phases. X-ray total scattering result also excluded the possibility of forming a small amorphous oxide phase in the hematite as a main product from the dopants (Fig. 3 and Supplementary Figs. 11 and 12).

Based on the reviewer's suggestion, we have measured X-ray absorption spectra in a CEY mode to analyse the compositions near the surface region of the sample. To compare with references and samples measured in the transmission mode, we adapted the k ranges of 1.0–6.8 \AA^{-1} and 1.0–9.0 \AA^{-1} for Ti K and Sn K EXAFS analyses, respectively. The Sn K-edge and Ti K-edge X-ray absorption spectra clearly show a transition from doped states to dopant oxide heterostructures by the annealing treatment. We added the updated X-ray absorption data in the revised manuscript as follows.

The Ti K-edge XANES spectrum line shape of both the as-synthesised Ti-Fe₂O₃ and SnTi-Fe₂O₃ is similar to that of the FeTiO₃ sample with the strongest third post-edge absorption at a similar position (Supplementary Figs. 15a and b), indicating the similar oxidation environment of Ti ions due to the doping of Ti⁴⁺ in hematite. For the annealed Ti-Fe₂O₃, a characteristic small pre-edge peak of rutile (as indicated by the arrow) was detected, and the line shape and absorption peaks in the post-edge region are also very close to that of rutile (Supplementary Figs. 15a and b), indicating the formation of rutile phase overlayer in this sample. This was further proven by the corresponding Ti K-edge EXAFS spectrum which shows the same Ti-coordination as that of rutile (Supplementary Figs. 15e and g). For the annealed SnTi-Fe₂O₃ sample, no characteristic small pre-edge peak of rutile was detected and the line shape and post-edge absorption peaks were also different from those of rutile (Supplementary Figs. 15b and d), suggesting the absence of rutile phase in this sample. In addition, the first post-edge absorption peak is becoming stronger and smoother compared to that of FeTiO₃, which is due to the formation of Sn–Ti coordination, as indicated by the corresponding Ti K-edge FT-EXAFS spectrum which shows a Sn–Ti shell located between that of Fe–Ti shell of FeTiO₃ and Sn–Sn shell of SnO₂ (Supplementary Figs. 15f and h).

Supplementary Fig. 15. **a**, Ti K-edge XANES spectra of as-synthesised Ti-Fe₂O₃ and annealed Ti-Fe₂O₃, and their corresponding FT-EXAFS spectra (**e**). **b**, Ti K-edge XANES spectra of as-synthesised SnTi-Fe₂O₃ and annealed SnTi-Fe₂O₃, and their corresponding FT-EXAFS spectra (**f**). **c**, Ti K-edge XANES spectra of annealed Ti-Fe₂O₃ measured in transmission and CEY modes, and their corresponding FT-EXAFS spectra (**g**). **d**, Ti K-edge XANES spectra of annealed SnTi-Fe₂O₃ measured in transmission and CEY modes, and their corresponding FT-EXAFS spectra (**h**).

The Sn K-edge EXAFS spectra of both the as-synthesised Sn-Fe₂O₃ and SnTi-Fe₂O₃ show the main shell at 2.76 Å (Supplementary Figs. 17e and f), which can be assigned to the Sn-Fe coordination due to the replacement of Fe³⁺ ions with Sn²⁺ ions in hematite (i.e. doping). The peak position of Sn-Sn for the annealed Sn-Fe₂O₃ is the same as that of the reference SnO₂ sample (Fig. 4d), proving the formation of a SnO₂ overlayer. For the annealed SnTi-Fe₂O₃, the Sn-Sn bonding has a shorter radial distance than that of SnO₂, which is due to the formation of Sn-Ti coordination (i.e., SnTiO_x) (Nanoscale 5, 2254 (2013); J. Colloid Interface Sci. 588, 242 (2021)).

Supplementary Fig. 17. a, Sn K-edge XANES spectra of as-synthesised Sn-Fe₂O₃ and annealed Sn-Fe₂O₃, and their corresponding FT-EXAFS spectra (e). b, Sn K-edge XANES spectra of as-synthesised SnTi-Fe₂O₃ and annealed SnTi-Fe₂O₃, and their corresponding FT-EXAFS spectra (f). c, Sn K-edge XANES spectra of annealed Sn-Fe₂O₃ measured in transmission mode and CEY mode, and their corresponding FT-EXAFS spectra (g). d, Sn K-edge XANES spectra of annealed SnTi-Fe₂O₃ measured in transmission and CEY modes, and their corresponding FT-EXAFS spectra (h).

3. The XPS shift of Sn and Ti may come from the wrong alignment of the XPS peak position. The two elements shift to the same energy direction with similar values.

Reply: Thanks for the reviewer's comment. We have carefully analysed the XPS data. The alignment of XPS curves was all calibrated by the C 1s peak (284.8 eV).

4. It is strange that the authors compared the XRD peak difference around 33 degree for the samples before annealing, while after annealing the XRD peaks are compared at around 24 degree. Both peaks should be compared before and after annealing.

Reply: Thanks to the reviewer for the suggestion. In the revised manuscript, we updated the XRD results (Fig. 3a, page 11).

5. Fig. 4a is not clear. So many curves are put together and it is hard to identify the difference. Moreover, the spectra of Ti-Fe₂O₃ and SnTi-Fe₂O₃ are almost identical to that of rutile TiO₂, suggesting the rutile structure. The white line peak position of Ti-Fe₂O₃ is almost the same as

that of rutile TiO₂.

Reply: Thanks for the reviewer's comments. Again we would like to explain the results of XAFS measurements. To identify the difference more clearly, we replaced Fig. 4a with one only showing the TiO₂, FeTiO₃, and the Ti-containing samples (Fig. 4a in the revised manuscript). For transition metals in oxides, the threshold energy position of the spectra is very sensitive to their oxidation states, while the shape of the peak gives information about the local structural environments of the absorbing elements. The line shapes of the annealed Ti-Fe₂O₃ MCs in pre-edge (with a characteristic peak of rutile as indicated by the arrow in Supplementary Fig. 15) and post-edge regions are similar to those of rutile TiO₂. For the annealed SnTi-Fe₂O₃ sample, the characteristic peak of rutile in the pre-edge (indicated by the black arrow) did not appear, and the peak shape and position were different from those of reference TiO₂ and annealed Ti-Fe₂O₃, indicating that no rutile TiO₂ phase was formed as the dominant product in this sample. The annealed SnTi-Fe₂O₃ sample also has a similar strongest absorption peak with FeTiO₃ at the post-edge region; thus, the oxidation state of Ti ions would be similar to FeTiO₃. In addition, a much stronger first post-edge peak may indicate the Sn–Ti coordination. This was further supported by the comparison of the ex situ Ti K edge XAS analysis measured in the transmission and CEY modes (Supplementary Fig. 15).

The Ti K edge XANES spectral line shape of both the as-synthesised Ti-Fe₂O₃ and SnTi-Fe₂O₃ samples is similar to that of FeTiO₃ with the strongest third post-edge absorption at a similar position (Supplementary Figs. 15a and b), indicating the similar oxidation environment of Ti ions due to the doping of Ti⁴⁺ in hematite. For the annealed Ti-Fe₂O₃, a characteristic small pre-edge peak of rutile (as indicated by the arrow) was detected, and the line shape and absorption peaks in the post-edge region are also very close to that of rutile (Supplementary Figs. 15a and b), indicating the formation of rutile phase overlayer in this sample. This was further proven by the corresponding Ti K edge EXAFS spectrum which shows the same Ti-coordination as that of rutile (Supplementary Figs. 15e and g). For the annealed SnTi-Fe₂O₃ sample, no characteristic small pre-edge peak of rutile was detected and the line shape and post-edge absorption peaks were also different from those of rutile (Supplementary Figs. 15b and d), suggesting the absence of rutile phase in this sample. In addition, the first post-edge absorption peak is becoming stronger and smoother compared to that of FeTiO₃, which is due to the formation of Sn–Ti coordination as indicated by the corresponding Ti K-edge FT EXAFS spectrum which shows a Sn–Ti shell located between that of Fe–Ti shell of FeTiO₃ and Sn–Sn shell of SnO₂ (Supplementary Fig. 15f and h).

6. In Fig. 4a the absence of pre-peak (the arrow) in the spectrum of SnTi-Fe₂O₃ might be related to the formation of amorphous TiO₂ with a small size. The different pre-peak position should be clarified based on the XANES spectral fitting or calculation. It cannot make a direct conclusion for the formation of FeTiO₃. It could also be Fe₂TiO₅. The in-situ spectral changes might be explained by the aggregation of small amorphous TiO₂ to rutile TiO₂.

Reply: We appreciate the reviewer's comments. As we responded earlier, all the experimental data indicated the conversion from doped Sn and Ti ions in the hematite lattice to their oxide hetero-overlayers as a result of dopant segregation followed by oxidation during the thermal treatment process. Fe₂TiO₅ is not considered as the main product according to the position of the white line peak (e.g., please see Catalysis Today 201, 131 (2013)). In addition, we were not able to obtain the experimental results (including in situ XANES), suggesting the formation of rutile TiO₂ for the annealed SnTi-Fe₂O₃. Since the present system contains the major hematite core (~100 nm) and very thin dopant oxide hetero-overlayers (1–7 nm thickness), it is difficult to accurately determine the local structures only from the XANES spectral fitting or calculation at present. However, we believe our experimental results in the revised manuscript are already strong enough to support our conclusion.

7. For the Sn K-edge EXAFS spectra, the SnO peak is too weak to confirm the existence of SnO. The formation of SnO or SnTiO_x will also change the first shell peak in the EXAFS spectra, but no clear evidence can be observed. The formation of SnTiO_x should be confirmed by the spectral fitting based on the corresponding models instead of a simple hypothesis. The present spectral changes are not a strong evidence for the formation of SnTiO_x. The Sn-Ti coordination cannot be proved in the whole manuscript.

Reply: We appreciate the reviewer for the comments and suggestions. Based on the reviewer's suggestion, we have measured X-ray absorption spectra in a CEY mode to analyse the compositions near the surface region of the prepared samples. To compare with references and samples measured in the transmission mode, we adapted the k ranges of 1.0–6.8 Å⁻¹ for Ti K and 1.0–9.0 Å⁻¹ for Sn K for EXAFS analysis. The Sn K-edge and Ti K-edge XANES and EXAFS clearly show a transition from doped states to dopant oxide heterostructures by the annealing treatment. After carefully analysing the obtained spectra, we can conclude the formation of Sn–Ti coordination. DFT calculations also gave insight into the surface structures (page 17).

We also agree with the reviewer's comments that the SnO peak is too weak to completely identify the in situ Sn K FT-EXAFS spectra. Because the phase transformation of SnO into SnO₂

occurs above 420 °C in air (*CrystEngComm* 16, 6841 (2014)), the spectrum at 450 °C would be a superposition of the spectra of SnO and SnO₂ (Fig. C4). Then, the Ti⁴⁺ ions that migrate to the surface react with SnO₂ to form SnTiO_x at the surface. Based on these considerations, we updated Figs. 4c and d and revised the manuscript as “Ex and in situ Sn K-edge FT-EXAFS measurements enhanced the investigation of the dynamics of dopant segregation (Figs. 2i and 4c, and Supplementary Figs. 17 and 18). The peak position of the Sn–Sn bond for annealed Sn-Fe₂O₃ sample is close to that of the reference SnO₂ (Fig. 2i, and Supplementary Fig. 17), proving the formation of SnO₂ overlayers. For the annealed SnTi-Fe₂O₃ sample, the second main shell has a shorter radial distance than that of SnO₂ owing to the formation of Sn–Ti coordination (i.e. SnTiO_x).^{29,30} The peak intensity of the Sn–Fe coordination shell gradually decreases as the temperature increases from 48 to 700 °C (Fig. 4c) because the Sn-coordination becomes disordered due to the diffusion of Sn ions from the hematite lattice. A second Sn–Sn coordination peak is formed and grows when the temperature rises to ~450 °C (Supplementary Fig. 19). Meanwhile, the Sn–Ti coordination peak appears at 700 °C and becomes stronger during the heat preservation and cooling stages, which agrees with the Ti K-edge XANES result (Fig. 4b). The growth of the Sn–Sn and Sn–Ti coordination peaks during the cooling stage indicates ordered Sn-coordination by suppressed oscillation or diffusion of the elements at lower temperatures. For Sn-Fe₂O₃ sample, the SnO₂ phase is formed, as indicated by the increase in the Sn–Sn coordination at ~550 °C (Supplementary Fig. 18). Based on these results, it is concluded that Sn²⁺ ions migrate from the hematite lattice before Ti⁴⁺ ions owing to their larger radius (i.e. larger EE) and segregate on the surface to form the SnO₂ (or SnO) phase at ~450 °C. The deficient SnTiO_x overlayers are then formed when Ti⁴⁺ ions segregate at the surface and react with SnO₂ at 700 °C (Fig. 4d). In addition, a very small amount of SnO₂ remained at the outer surface during the annealing treatment at 700 °C.” (page 14).

Fig. C4 is given as Supplementary Fig. 19 with the above discussion.

Fig. C4. Sn K-edge FT-EXAFS spectra obtained from in situ measurements of SnTi-Fe₂O₃ (see Fig. 4c). The spectra of the reference samples are also shown.

8. The stable time in Fig. S18a is too short. It should be at least more than 1 hour for a stable catalyst.

Reply: We thanks the reviewer for their suggestion. We have added a long-time stability test result in the revised manuscript (Supplementary Fig. 22d), which shows good stability.

REVIEWER COMMENTS

Reviewer #1 (Remarks to the Author):

The authors have successfully addressed our comments and the article is ready for publication.

Reviewer #3 (Remarks to the Author):

The authors have addressed all my questions. It can be published now.

Reviewer #4 (Remarks to the Author):

In the manuscript, the authors performed an excellent job on the synthesis of hematite-based heterostructure photoanode materials through dopant segregation. The SnTi-Fe₂O₃ heterostructure photoanode could efficiently oxidizes water to H₂O₂ with nearly 100% faraday efficiency. Some insights were also provided in the manuscript for the synthesis method. They have well addressed most of the question proposed by the reviewers. However, the DFT calculation results provided in the revised manuscript were questionable, which may not be able to convince the mechanism:

1. Fe₂O₃ is a strongly correlated material. The DFT+U or hybrid functional calculation should be performed rather than just pure GGA functional.

2. According to their experiment results, no rutile phase could be found in the overlayer of the best annealed SnTi-Fe₂O₃ photoanode for H₂O₂ formation. Hence, the used rutile model of Ti-SnO₂ or Sn-TiO₂ could not confirm the mechanism.

3. In addition, the performance of photoanode towards the production of H₂O₂ does be related to the $\Delta G(\text{OH}^*)$. However, another curve of limiting potential against ΔG_{OH^*} for the 4e process to produce O₂ should be also presented to confirm the high selectivity towards H₂O₂.

Responses to the comments from Reviewer #1:

The authors have successfully addressed our comments and the article is ready for publication.

Reply: Thank you very much for your recommendation.

Responses to the comments from Reviewer #3:

The authors have addressed all my questions. It can be published now.

Reply: Thank you very much for your recommendation.

Responses to the comments from Reviewer #4:

In the manuscript, the authors performed an excellent job on the synthesis of hematite-based heterostructure photoanode materials through dopant segregation. The SnTi-Fe₂O₃ heterostructure photoanode could efficiently oxidizes water to H₂O₂ with nearly 100% faraday efficiency. Some insights were also provided in the manuscript for the synthesis method. They have well addressed most of the question proposed by the reviewers. However, the DFT calculation results provided in the revised manuscript were questionable, which may not be able to convince the mechanism:

Reply: We appreciate the reviewer for the constructive comments and suggestions.

1. Fe₂O₃ is a strongly correlated material. The DFT+U or hybrid functional calculation should be performed rather than just pure GGA functional.

Reply: We thank the reviewer for this suggestion. We have carried out calculations using RPBE + U ($U = 4.3$ eV) changing the surface coverage. In the original manuscript, we only showed the results for the 1/2 ML coverage, but this choice might have been inappropriate as it turned out that ΔG_{*OH} is highly correlated with surface coverage as discussed below.

For 1/12 ML coverage (Terminal site), the computed ΔG_{*OH} values were 0.62 and 1.33 eV, without and with U , respectively. This increase is in fact the same trend as has been discussed in Xu et al., J. Phys. Chem. C 119, 4827 (2015); namely, introducing the Hubbard correction generally weakens adsorption. Furthermore, the latter value is also in good agreement with the previously reported value, approximately 1.5 eV, by Zhang et al., J. Phys. Chem. C 120, 18201 (2016) (see Figure 3), which was obtained by using PBE + U ($U = 4.3$ eV) and 1/12 ML.

With $\Delta G_{*O} = 3.45$ eV computed with U (1.96 eV without U), which also agrees well with the above reference, the O₂ evolution reaction is predicted to dominate over the H₂O₂ evolution reaction in Fe₂O₃.

As soon as the coverage increases from 1/12 ML, we found a large increase in ΔG_{OH^*} with $U = 4.3$ eV to 1.74–1.88 eV on average. The computed ΔG_{OH^*} for 2/12 ML was found to be 1.74 eV on average, indicating a free energy of 2.15 eV is required to further attach the second OH to the 1/12 ML surface. This is an important insight, because it is strongly indicated that this step has a large thermodynamic barrier and is unlikely to occur at an extra bias of 1.76 eV, ideal for the H_2O_2 evolution reaction. These results also support the experimental evidence that Fe_2O_3 is an O_2 evolution catalyst.

To address these results, we summarized the RPBE+U calculations and briefly discussed the coverage dependence in the Supplementary Information (Supplementary Note 3, Supplementary Fig. 25, and Supplementary Table 2).

2. According to their experiment results, no rutile phase could be found in the overlayer of the best annealed SnTi-Fe₂O₃ photoanode for H₂O₂ formation. Hence, the used rutile model of Ti-SnO₂ or Sn-TiO₂ could not confirm the mechanism.

Reply: We thank the reviewer for this suggestion. As the reviewer mentioned, no rutile phase was observed for the annealed SnTi-Fe₂O₃. Since the surface overlayers on the annealed SnTi-Fe₂O₃ are structurally disordered (not well-defined crystal structures), we modelled rutile Ti-SnO₂ and Sn-TiO₂ as the local structures. From the calculation results of Ti-SnO₂ and Sn-TiO₂, we found that a simple doping treatment cannot improve the selectivity for H₂O₂ evolution. To clarify this point, we revised the manuscript (page 17) as “We also point out that a simple doping treatment cannot improve the reaction selectivity, according to the fact that the ΔG_{OH^*} values calculated for two local structures, Sn⁴⁺-doped TiO₂ (Sn-TiO₂) and Ti⁴⁺-doped SnO₂ (Ti-SnO₂), where the dopants are considered as the surface active sites, are comparable to those of SnO₂ and TiO₂, respectively (Fig. 5f, Supplementary Fig. 26, and Supplementary Table 3).”.

3. In addition, the performance of photoanode towards the production of H₂O₂ does be related to the $\Delta G(\text{OH}^*)$. However, another curve of limiting potential against ΔG_{OH^*} for the 4e process to produce O₂ should be also presented to confirm the high selectivity towards H₂O₂.

Reply: We thank the reviewer for this suggestion. We showed the product selectivity diagram in terms of ΔG_{OH^*} and ΔG_{O^*} in Supplementary Fig. 26. This diagram indicates the two regions in which O₂ and H₂O₂ are expected to be the major product on the basis of purely thermodynamic considerations. According to the criteria given by Nørskov and co-workers (Nat. Commun. 8, 701 (2017); J. Phys. Chem. Lett. 8, 1157 (2017)), H₂O₂ synthesis is favoured when ΔG_{OH^*} is between

1.6 and 2.4 eV and ΔG_{O^*} is larger than 3.5 eV.

Supplementary Fig. 26. Product selectivity diagram in terms of ΔG_{OH^*} and ΔG_{O^*} . Blue and red highlighted colours indicate the regions in which O₂ and H₂O₂ are expected to be the major product, respectively, on the basis of purely thermodynamic considerations. H₂O₂ synthesis is favoured when ΔG_{OH^*} is between 1.6 and 2.4 eV and ΔG_{O^*} is larger than 3.5 eV.^{15,28}

REVIEWER COMMENTS

Reviewer #4 (Remarks to the Author):

The authors have well addressed my previous questions of 1 and 3. Still, they did not provide the theoretical insight into the best performance of the annealed SnTi-Fe₂O₃ photoanode for H₂O₂ formation. The DFT calculation results should be added for this complex structure to confirm their conclusion.

Responses to the comments from Reviewer #4:

The authors have well addressed my previous questions of 1 and 3. Still, they did not provide the theoretical insight into the best performance of the annealed SnTi-Fe₂O₃ photoanode for H₂O₂ formation. The DFT calculation results should be added for this complex structure to confirm their conclusion.

Reply: We appreciate the reviewer for the comment. In line with the comment, we carried out calculations for SnTiO₃ and Sn_{0.5}Ti_{0.5}O₂, which are possible SnTiO_x structures, as follows.

Recently, Diehl et al. reported an ilmenite-type SnTiO₃ structure (Chem. Mater. 30, 8932 (2018); Chem. Mater. 33, 2824 (2021)). However, our calculations revealed that ideal SnTiO₃ (0001) surface has a ΔG_{OH^*} of 1.24 eV (ΔG_{O^*} of 3.04 eV), which is not suitable for H₂O₂ evolution. These results were given in Fig. 5f, Supplementary Figs. 26 and 38, and Supplementary Table 3. They also reported that an oxidised passivation layer (~2 nm thickness) that resembles SnO₂ formed at the top surface of SnTiO₃. This is the reason why we selected rutile SnO₂ with V_O as a local structure of catalytic sites on the annealed SnTi-Fe₂O₃. When V_O was introduced near the surface of SnO₂, the OH adsorption was significantly enhanced, leading to a poor H₂O₂ evolution activity (e.g., #1 in Figs. 5e and f). Meanwhile, when V_O is present in deeper positions, the ΔG_{OH^*} value shifts toward the volcano peak where the catalyst is optimal for H₂O₂ production. The subsurface V_O mimics the Sn²⁺ support from SnTiO₃. We further examined rutile-type Sn_{0.5}Ti_{0.5}O_{2-x} (Chen et al., Nanoscale 5, 2254 (2013)) and found a similar tendency as in SnO_{2-x}, as shown in Supplementary Fig. 30 and Supplementary Table 5. Since the surface Sn²⁺ ions are probably oxidised to Sn⁴⁺ during the annealing treatment in air (Supplementary Fig. 8d), SnO₂ (or Sn_{0.5}Ti_{0.5}O₂) (below 2 nm thickness) could form at the outer surface of disordered SnTiO_x overlayers. At the present stage, we think this structure is a possible catalytic site on the SnTi-Fe₂O₃ for efficient H₂O₂ production.

We added the following sentences in the revised manuscript. Thank you very much again for your constructive comment.

On page 17: Recently, Diehl et al. reported an ilmenite-type SnTiO₃ structure where each Sn²⁺ possesses a lone pair, forming layers separated by a van der Waals gap.^{45,46} This finding inspired us to explore local structures of SnTiO_x overlayers, but our calculations revealed that ideal SnTiO₃ (0001) surface has a ΔG_{OH^*} of 1.24 eV (ΔG_{O^*} of 3.04 eV), which is not suitable for H₂O₂ evolution (Fig. 5f, Supplementary Fig. 28, and Supplementary Table 3). They also reported that an oxidised passivation layer (~2 nm thickness) that resembles SnO₂ formed at the top surface of SnTiO₃. We

thus modeled various rutile SnO_2 structures possessing V_O , where subsurface V_O mimics the Sn^{2+} support from SnTiO_3 , and examined their possibility as a catalytic site on the annealed $\text{SnTi-Fe}_2\text{O}_3$. As demonstrated in Figs. 5e and f, when V_O was introduced near the surface of SnO_2 (site #1), the OH adsorption was significantly enhanced, leading to a poor H_2O_2 evolution activity.

On page 18: We also found a similar tendency for rutile-type $\text{Sn}_{0.5}\text{Ti}_{0.5}\text{O}_{2-x}$ with V_O (Supplementary Fig. 30 and Supplementary Table 5).³⁰ ... Since the surface Sn^{2+} ions are probably oxidised to Sn^{4+} during the annealing treatment in air (Supplementary Fig. 8d), partially amorphised SnO_2 (or $\text{Sn}_{0.5}\text{Ti}_{0.5}\text{O}_2$) (below 2 nm thickness) could form at the outer surface of disordered SnTiO_x overlayers. Such a heterostructure could be realised by successive binary dopant segregation through nanoparticle networks in the MCs (Fig. 4d). Among the structures utilised in the DFT calculations, the prospective ones are SnO_{2-x} or $\text{Sn}_{0.5}\text{Ti}_{0.5}\text{O}_{2-x}$ with V_O at depths of 1.2–1.7 nm (e.g., site #5 in Figs. 5e and f, and Supplementary Fig. 29), which are structurally analogous to the Sn^{4+} species on the disordered SrTiO_x overlayers.

REVIEWERS' COMMENTS

Reviewer #4 (Remarks to the Author):

The authors have well addressed my question. I suggest the manuscript be publishable in Nature Communications.

Responses to the comment from Reviewer #4:

The authors have well addressed my question. I suggest the manuscript be publishable in Nature Communications.

Reply: Thank you very much for your recommendation.